EMBO
Molecular Medicine

# FABP4 as a therapeutic host target controlling SARS-CoV-2 infection

Hatoon Baazim[1], Emre Koyuncu [2], Gürol Tuncman [1], M Furkan Burak[1,3], Lea Merkel [1], Nadine Bahour[1], Ezgi Simay Karabulut[1], Grace Yankun Lee [1], Alireza Hanifehnezhad [4], Zehra Firat Karagoz [5], Katalin Földes[6], Ilayda Engin[5], Ayse Gokce Erman[5], Sidika Oztop [7], Nazlican Filazi [8], Buket Gul [4], Ahmet Ceylan[9], Ozge Ozgenc Cinar[9], Fusun Can[10], Hahn Kim[2,11,12], Ali Al-Hakeem [2], Hui Li[13], Fatih Semerci [2], Xihong Lin [13], Erkan Yilmaz[5], Onder Ergonul [10], Aykut Ozkul [4✉] & Gökhan S Hotamisligil [1,14✉]

## Abstract

**Host metabolic fitness is a critical determinant of infectious disease outcomes. Obesity, aging, and other related metabolic disorders are recognized as high-risk disease modifiers for respiratory infections, including coronavirus infections, though the underlying mechanisms remain unknown. Our study highlights fatty acid-binding protein 4 (FABP4), a key regulator of metabolic dysfunction and inflammation, as a modulator of SARS-CoV-2 pathogenesis, correlating strongly with disease severity in COVID-19 patients. We demonstrate that loss of FABP4 function, by genetic or pharmacological means, reduces SARS-CoV-2 replication and disrupts the formation of viral replication organelles in adipocytes and airway epithelial cells. Importantly, FABP4 inhibitor treatment of infected hamsters diminished lung viral titers, alleviated lung damage and reduced collagen deposition. These findings highlight the therapeutic potential of targeting host metabolism in limiting coronavirus replication and mitigating the pathogenesis of infection.**

**Keywords** FABP4; SARS-CoV-2; COVID-19; Replication Organelles; Lipid Droplets
**Subject Categories** Metabolism; Microbiology, Virology & Host Pathogen Interaction

## Introduction

Viral infections are highly disruptive events that mobilize extensive resources to facilitate replication, organelle remodeling (Cortese et al, 2020; Roingeard et al, 2022) and immune responses (Diamond and Kanneganti, 2022). The resulting immunometabolic interactions substantially modify host metabolism, while also being greatly influenced by it (Lercher et al, 2020). The importance of the host's metabolic state in determining infectious disease outcomes has been well established, particularly in conditions such as obesity, diabetes, and aging, which substantially increase morbidity and mortality during coronavirus infections even in vaccinated individuals (Fan et al, 2023; Yang et al, 2017). However, a mechanistic understanding of how metabolism affects viral replication and pathogenesis remains limited. In the context of obesity, current hypotheses attribute this heightened pathogenesis to the underlying metabolic dysregulation and low-grade inflammation (Shaikh et al, 2024; Hotamisligil, 2017), which could impair anti-viral immune responses and cause airway hyperresponsiveness (Scott et al, 2023). It is also possible that ectopic lipid deposition and excess nutrient flux, resulting from uncontrolled adipose tissue lipolysis and insulin resistance (Schweiger et al, 2017), could create an environment that is either more conducive to viral replication or more susceptible to tissue damage, due to lipotoxicity and oxidative stress. Studies implementing mouse-adapted viral strains of the Middle East respiratory syndrome coronavirus (MERS-CoV) and SARS-CoV-2 reported increased mortality in obese mice despite comparable lung viral titers (Johnson et al, 2023; Kulcsar et al, 2019).

FABP4 is an exceptionally versatile regulator of energy resources (Prentice et al, 2021, 2019) and a modulator of metabolic and inflammatory responses (Ge et al, 2018; Shum et al, 2006). Our

[1]Sabri Ülker Center for Metabolic Research, Department of Molecular Metabolism, Harvard T.H. Chan School of Public Health, Boston, MA, USA. [2]Crescenta Biosciences Inc, Irvine, CA, USA. [3]Division of Endocrinology, Diabetes and Hypertension, Brigham and Women's Hospital, Harvard Medical School, Boston, MA, USA. [4]Ankara University, Faculty of Veterinary Medicine, Department of Virology, Ankara, Türkiye. [5]Ankara University, Biotechnology Institute, Ankara, Türkiye. [6]The Pirbright Institute, Woking, UK. [7]Ankara Medipol University, School of Medicine, Department of Medical Biology, Ankara, Türkiye. [8]Mustafa Kemal University, Faculty of Veterinary Medicine, Department of Virology, Hatay, Türkiye. [9]Ankara University, Faculty of Veterinary Medicine, Department of Histology and Embryology, Ankara, Türkiye. [10]Koç University, School of Medicine, Department of Infectious Diseases, Istanbul, Türkiye. [11]Princeton University Small Molecule Screening Center, Princeton University, Princeton, NJ, USA. [12]Department of Chemistry, Princeton University, Princeton, NJ, USA. [13]Department of Biostatistics, Harvard T.H. Chan School of Public Health, Boston, MA, USA. [14]Harvard-MIT Broad Institute, Cambridge, MA, USA. ✉E-mail: Aykut.Ozkul@ankara.edu.tr; ghotamis@hsph.harvard.edu

current understanding of FABP4 biology attributes its versatility to its dual function, as an intracellular lipid chaperon, delivering a wide range of ligands to phospholipid membranes or lipid-sensing receptors such as PPARγ (Gericke et al, 1997; Tan et al, 2002), as well as a secreted hormone capable of interacting with other proteins, such as NDPK and ADK (Prentice et al, 2021). While there is no known direct link between FABP4 and coronavirus infections, FABP4 is known to promote inflammation and metabolic dysfunction in several comorbidities that constitute a high-risk for coronavirus infection, including diabetes (Prentice et al, 2021), cardiovascular disease (Erbay et al, 2009), and airway disease (Ge et al, 2018; Shum et al, 2006). In addition, its regulation of de novo lipogenesis (Garin-Shkolnik et al, 2014), lipid composition (Cao et al, 2008) and intracellular lipid trafficking makes it a promising candidate in facilitating SARS-CoV-2 induced organelle remodeling and lipid droplet (LD) utilization (Dias et al, 2020). Adipose tissue dysfunction is another common link amongst COVID-19 high-risk comorbidities through its regulation of systemic metabolism and its ability to secrete proinflammatory cytokines (Shaikh et al, 2024). Adipocytes are permissible to SARS-CoV-2 infection (Saccon, 2022; Zickler et al, 2022), and the detection of viral RNA in the adipose tissue of deceased COVID-19 patients and infected animals (Saccon, 2022; Zickler et al, 2022; Martínez-Colón et al, 2022; Reiterer et al, 2021) indicates that viral dissemination into the adipose tissue may occur naturally. Adipocytes are the most abundant source of FABP4 (Matarese and Bernlohr, 1988), and together with macrophages, endothelial and epithelial cells, present an overlapping target for SARS-CoV-2 infection and FABP4 action. These observations prompted us to directly assess the impact of FABP4 in COVID-19.

In this study, we observed significantly elevated FABP4 levels in the serum and lungs of COVID-19 patients that highly correlate with disease severity. We then examined the intracellular interaction between FABP4 and SARS-CoV-2 using cultured adipocytes, and bronchial epithelial cells. We demonstrated that, FABP4 is recruited to the double-membrane vesicles (DMVs) of virus replication organelles (ROs), which are membrane-bound compartments derived from remodeled cellular organelles (Cortese et al, 2020; Roingeard et al, 2022). Pharmacological inhibition, or genetic deletion of FABP4 resulted in disruption of DMV numbers and organization in SARS-CoV-2 infected cells, and broadly impaired coronavirus propagation across various cell types infected with different SARS-CoV-2 strains and the common cold coronavirus OC43 (HCoV-OC43). Importantly, FABP4 inhibition in infected hamsters significantly reduced viral titers, and ameliorated lung damage and fibrosis. Together, these data indicate that FABP4 facilitates SARS-CoV-2 infection and that therapeutic targeting of FABP4 may be a beneficial treatment strategy for mitigating COVID-19 severity and mortality in humans.

## Results

### FABP4 protein levels correlate with disease severity in COVID-19 patients

To assess FABP4's involvement in SARS-CoV-2 infection, we conducted immunohistochemical staining for FABP4 in lung biopsies obtained from individuals with COVID-19. This revealed a high FABP4 signal in the endothelial cells lining the small vessels of the lung parenchyma (Fig. 1A). We also examined circulating levels of FABP4 in two patient cohorts at distinct stages of disease severity defined according to the WHO severity criteria and 45 healthy controls. [Cohort 1: November 2020–May 2021, $n = 283$, dominant variants: epsilon and alpha (Peacock et al, 2021). Cohort 2: March–May 2020, $n = 116$, dominant variant: original Wuhan strain] (Tables 1–3, Table EV1, Datasets EV1 and EV2). Multiple serum samples were collected from each patient during hospitalization and marked according to the day post-symptom onset. This analysis revealed a progressive increase in FABP4 levels (Figs. 1B,C and EV1A) that corresponded with an increase in several biomarkers of disease severity, including IL-6, C-reactive protein and circulating leukocytes (Figs. 1D and EV1B–D) and a decrease in lymphocytes (Fig. EV1E). We further interrogated this association by performing a regression analysis that examines FABP4 concentration over time in each patient, adjusting for age, sex, and BMI. This analysis uncovered a strong correlation between increased circulating FABP4 levels and COVID-19 disease severity (Fig. 1E). Consistent with these results, we found that higher FABP4 levels also correlated with the need for non-invasive or mechanical ventilation (Fig. 1F). We then examined FABP4 levels in severe and critically ill patients across several high-risk disease modifiers where FABP4 is known to promote inflammation and metabolic dysfunction. By doing so, we found that patients with reported comorbidities and those over 50 years of age had elevated FABP4 levels (Figs. 1G,H and EV1F,G). Similar results were also found in patients with a BMI equal to or over 30 and those with cardiometabolic conditions (Fig. EV1H,I). Taken together, this data points to a strong regulation and potential involvement of FABP4 in the pathophysiology of SARS-CoV-2 and suggests that the presence of underlying conditions that are regulated by FABP4 could further influence its involvement.

### SARS-CoV-2 infection is accelerated in differentiated adipocytes where FABP4 is recruited to DMVs

Due to the central role of adipocytes in aging and obesity-related metabolic dysfunction, and their high FABP4 abundance (Matarese and Bernlohr, 1988), we investigated the dynamics of SARS-CoV-2 infection using two infection titers (MOI = 0.1 or 1, SARS-CoV-2 WA1/2020), in the human Telomerase Reverse Transcriptase (hTERT) immortalized preadipocyte cell lines (Lee et al, 2004), before and after differentiation. Interestingly, in both infection titers, viral RNA levels were significantly higher in differentiated adipocytes, and viral titers, measured by plaque assay, exceeded those of pre-adipocytes by several orders of magnitude (Figs. 2A–C and EV2A–C). A closer examination of the infection in differentiated adipocytes revealed that while viral RNA levels were seemingly unchanged over time, nucleocapsid protein abundance did increase (Fig. EV2E,I). Moreover, infection of differentiated adipocytes triggered a marked increase in IL-6 secretion, at concentrations proportional to the inoculation titers (Figs. 2D and EV2D). Adipocytes infected at various points during their differentiation, with readouts taken 48 h post-infection, revealed that more differentiated adipocytes had a markedly higher capacity for viral replication, as evident by the increased virus titers and nucleocapsid protein abundance (Fig. 2E–G). As expected, FABP4 abundance also increased during adipocyte differentiation (Schlottmann et al, 2014) (Fig. 2F,G).

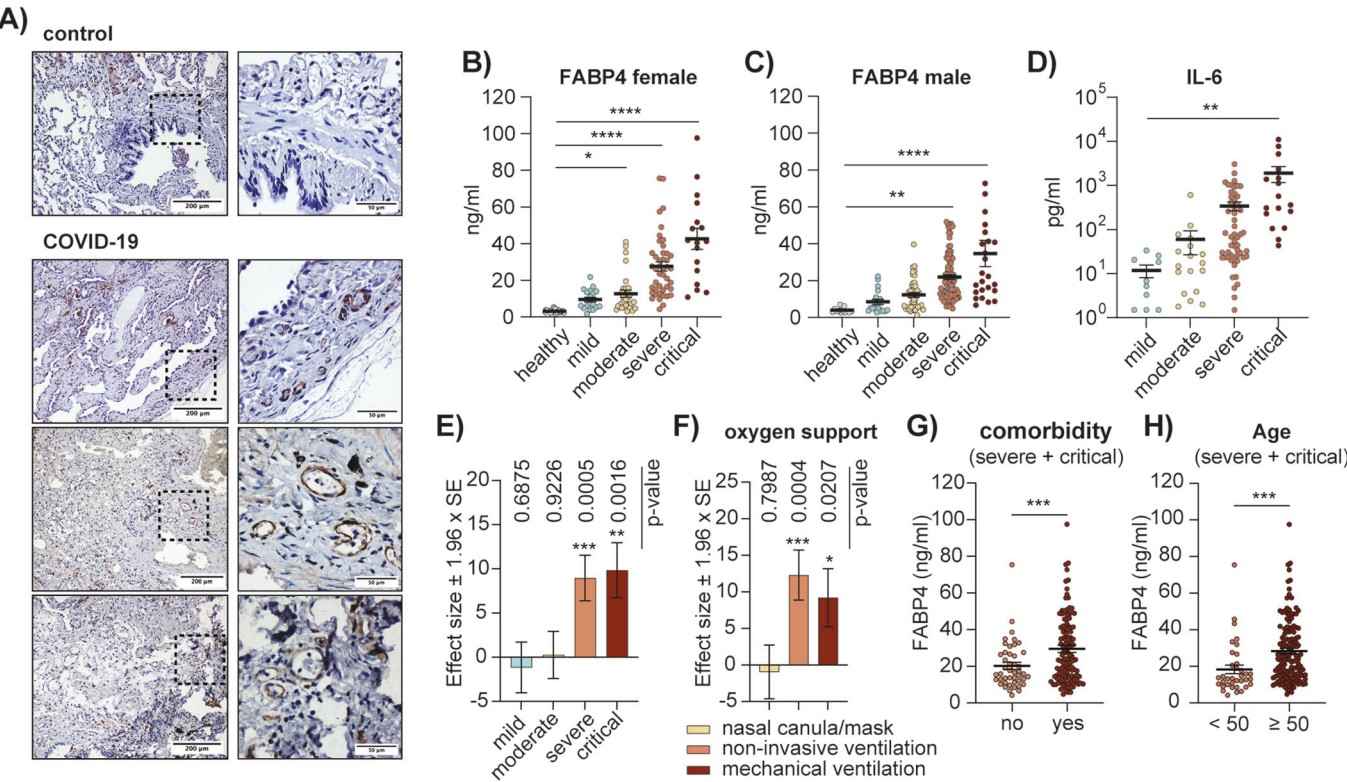

**Figure 1. FABP4 levels are increased in the lungs and circulation of COVID-19 patients.**

(A) Immunohistochemical staining of lung biopsies of controls and COVID-19 patients, using anti-FABP4 antibody. (B, C) Circulating FABP4 concentrations measured from (B) female ($n = 109$ female patients) and (C) male COVID-19 patients ($n = 174$ male patients) and healthy controls ($n = 32$ female and $n = 13$ male), stratified based on disease severity (****$p < 0.0001$, **$p = 0.002$, *$p = 0.0221$). (D) Circulating IL-6 levels in COVID-19 patients, stratified based on disease severity (**$p = 0.0017$). (B–D) Statistical analysis was performed using one-way ANOVA. (E, F) Effect size estimates and inference based on regression analysis of FABP4 concentration on (E) COVID-19 severity and (F) oxygen support measures while accounting for time of collection post symptom onset, age, sex, and BMI, using the linear mixed model to account for the patient-level random effects. (E) The healthy controls and, (F) patients who did not require oxygen support were used as a reference group. p-values are calculated based on the Wald test. For (B–F), the analysis included $n = 283$ total patients and $n = 45$ healthy controls. (G, H) FABP4 concentration in patients with severe and critical disease ($n = 176$ patients), stratified based on (G) presence or absence of comorbidities (listed in Table 2 and Dataset EV1, ***$p = 0.0009$) or (H) age (***$p = 0.0006$). Statistical analysis was performed using Welche's t-test. (B–H) Data are derived from patient cohort 1 (collected November 2020–May 2021). Patients were sampled longitudinally, and the data shown in (B), (C), (D), (G) and (H) represent the maximum measured concentration per patient. Data are shown as the mean ± s.e.m. Source data are available online for this figure.

**Table 1. Healthy controls.**

|  | Total ($n = 45$) | Female ($n = 32$) | Male ($n = 13$) |
|---|---|---|---|
| Median Age | 28 (20–56) | 27 (20–56) | 28 (22–41) |
| Median BMI | 21.9 (17–30) | 21 (17–28) | 24.6 (21–30) |

Cohort 1 (November 2020–May 2021).

To further examine the effect of infection on adipocytes' lipid stores, we examined the size of lipid droplets (LDs) in infected cells, identified using either the nucleocapsid protein or the double-stranded RNA (dsRNA), a product of viral replication detected at early stages of infection (Cortese et al, 2020). We found that overall, the LDs of infected cells were significantly smaller compared to neighboring cells (Fig. 2H,I). Notably, the size of the LDs in dsRNA-positive cells had a much wider variation compared to those in nucleocapsid-positive cells and showed a negative correlation with the dsRNA signal but not with the nucleocapsid signal (Figs. 2J,K and EV2K). This correlation was clearly observable in our confocal images, where cells with fewer dsRNA puncti had larger LDs and vice versa (Fig. 2J). Overall, these

results confirm that hTERT cells provide a good model to study SARS-CoV-2 infection in human adipocytes and suggest that SARS-CoV-2 replication is accelerated in cells of a higher lipid content, and that infection results in a gradual depletion of the cell's lipid stores.

To contextualize the role of intracellular FABP4 during infection, we examined its protein abundance and RNA expression but found no changes across the samples collect 24 and 48 h after infection (Fig. EV2E–G). We then measured FABP4 secretion at 12, 24, and 48 h post infection by incubating the cells with a fresh low-volume media for 1 h at each of the indicated time points. Compared to uninfected cells, FABP4 secretion from infected cells was marginally reduced within the first 12 h post infection (Fig. EV2H). Fluorescence staining revealed a striking change in the spatial distribution of FABP4 within infected cells. FABP4, typically a cytosolic protein, condensed into puncti that co-localized with the dsRNA and ER membrane protein calnexin (Fig. 2L–P), indicating its recruitment to the DMVs of the virus ROs (Knoops et al, 2008; Hackstadt et al, 2021). Whereas in nucleocapsid positive cells, the FABP4 signal was distributed across

**Table 2. COVID-19 patients.**

| | | COVID-19 disease severity | | | |
| | Total (%) | Mild (%) | Moderate (%) | Severe (%) | Critical (%) |
|---|---|---|---|---|---|
| Total patients | 283 | 35 (12.4) | 72 (25.4) | 134 (47.3) | 42 (14.8) |
| Female | 109 (38.5) | 18 (6.4) | 30 (10.6) | 44 (15.5) | 17 (6) |
| Male | 174 (61.5) | 17 (6) | 42 (14.8) | 90 (31.8) | 25 (8.8) |
| Median Age | 61 (21– 99) | 53 (21–87) | 60 (24–83) | 59 (33–99) | 74 (54–93) |
| Median BMI | 27.2 (17.5–48.5) | 26 (17.5–36.1) | 26.2 (20–48.4) | 26.8 (22.6–38.5) | 25.9 (24.6–39.7) |
| ICU admission | 47 (16.6) | 0 (0.0) | 0 (0.0) | 6 (2.1) | 41 (14.5) |
| Mortality | 14 (4.9) | 0 (0.0) | 0 (0.0) | 0 (0.0) | 14 (4.9) |
| All Comorbidity | 210 (74.2) | 26 (9.2) | 54 (19.1) | 93 (32.9) | 39 (13.8) |
| Hypertension | 95 (44.8) | 8 (2.8) | 32 (11.3) | 42 (14.8) | 21 (7.4) |
| Coronary Artery Disease | 30 (14.2) | 2 (0.7) | 3 (1.1) | 16 (5.7) | 9 (3.2) |
| Obesity | 7 (3.3) | 0 (0.0) | 2 (0.7) | 2 (0.7) | 3 (1.1) |
| Diabetes | 57 (26.9) | 6 (2.1) | 14 (4.9) | 29 (10.2) | 8 (2.8) |
| COPD | 11 (5.2) | 1 (0.4) | 0 (0.0) | 4 (1.4) | 6 (2.1) |
| Asthma | 9 (4.2) | 2 (0.7) | 2 (0.7) | 5 (1.8) | 0 (0.0) |

Cohort 1 (November 2020–May 2021).

the cytoplasm, showing a more heterogenous pattern of distribution (Fig. EV2J). Notably, both patterns were observed in cells neighboring nucleocapsid positive cells, suggesting a dynamic change in FABP4 distribution at different stages of infection. Indeed, when examining the co-localization scores over time, we found that FABP4's association with dsRNA and calnexin occurred as early as 8 h post infection, and peaked at 12 to 24 h (Fig. 2N,P). The co-localization between calnexin and dsRNA on the other hand was at its peak 48 h post infection (Fig. 2O). This data show that FABP4 is recruited to DMVs at the early stages of virus RO biogenesis and may play a role in virus replication.

## FABP4 deficiency impairs coronavirus replication

To understand the functional relevance of FABP4 in SARS-CoV-2 infection, we utilized small molecule inhibitors of FABP4. For these experiments, we used two structurally different inhibitors targeting FABP4: BMS309403 (Furuhashi et al, 2007; Shinoda et al, 2020) and CRE-14 (Koyuncu et al, 2023), which represents a newly developed class of FABP inhibitors. We validated and confirmed the activity of these molecules in competing with FABP4 ligands using the Terbium-based time-resolved fluorescence energy transfer assay (TR-FRET), which confirmed that both BMS309403 and CRE-14 were able to block FABP4's ability to bind its lipid ligand (BODIPY FL C12) (Fig. EV3A). BMS309403 has been reported to bind to FABP4 with a $K_D$ of 552 nM (Shinoda et al, 2020). Using a micro-scale thermophoresis (MST) assay, we determined that CRE-14 binds FABP4 with a $K_D = 954$ nM (Fig. EV3B,C). Finally, we examined the effect of CRE-14 treatment on cell viability by treating MRC5 cells with increasing concentrations of the inhibitor (Fig. EV3D).

Differentiated human adipocytes were treated with FABP4 inhibitors following SARS-CoV-2 infection (MOI = 0.1 or 1) at a high dosage (20 μM). This inhibition significantly reduced viral nucleocapsid RNA expression and protein abundance, leading to a

**Table 3. COVID-19 patients.**

| | | | COVID-19 disease severity | |
| | Total | Moderate | Severe | Critical |
|---|---|---|---|---|
| Total patients | 116 | 52 (44.8) | 21 (18.1) | 42 (36.2) |
| Female | 48 (41.4) | 29 (25) | 7 (6) | 12 (10.3) |
| Male | 68 (58.6) | 23 (19.8) | 14 (12.1) | 31 (26.7) |
| Median Age | 65 (35–91) | 63 (18–87) | 60 (35–77) | 69 (45–91) |
| Median BMI | 27.4 (19.6–55) | 27.6 (18.6–55) | 27 (22–44) | 28 (19.6–52.1) |
| ICU admission | 31 (26.7) | 0 (0.0) | 0 (0.0) | 31 (26.7) |
| Mortality | 9 (7.8) | 0 (0.0) | 0 (0.0) | 9 (7.8) |
| All Comorbidity | 58 (50) | 25 (21.6) | 7 (6) | 26 (22.4) |
| Hypertension | 46 (39.7) | 22 (19) | 6 (5.2) | 18 (15.5) |
| Coronary Artery Disease | 20 (17.2) | 9 (7.8) | 4 (3.4) | 7 (6) |
| Obesity | 8 (6.9) | 1 (0.9) | 1 (0.9) | 6 (5.2) |
| DM | 25 (21.6) | 8 (3.4) | 4 (3.4) | 13 (11.2) |
| COPD | 1 (0.9) | 0 | 0 | 1 (0.9) |
| Asthma | 2 (1.7) | 0 | 0 | 2 (1.7) |

Cohort 2 (March–May 2020).

marked reduction in viral titers (Figs. 3A–D and EV3E–H). In addition, FABP4 inhibition significantly decreased IL-6 secretion 48 h post-infection (Fig. 3E), suggesting an improved inflammatory state. To further confirm the relevance of FABP4 in SARS-CoV-2 replication in a genetic model, we generated FABP4-deficient human adipocyte cell lines through CRISPR-mediated deletion (Fig. 3F). We validated that this methodology was successful in eliminating FABP4 in these adipocytes, as shown by western blot analysis (Fig. 3F), and confirmed that FABP4$^{-/-}$ cells retain their

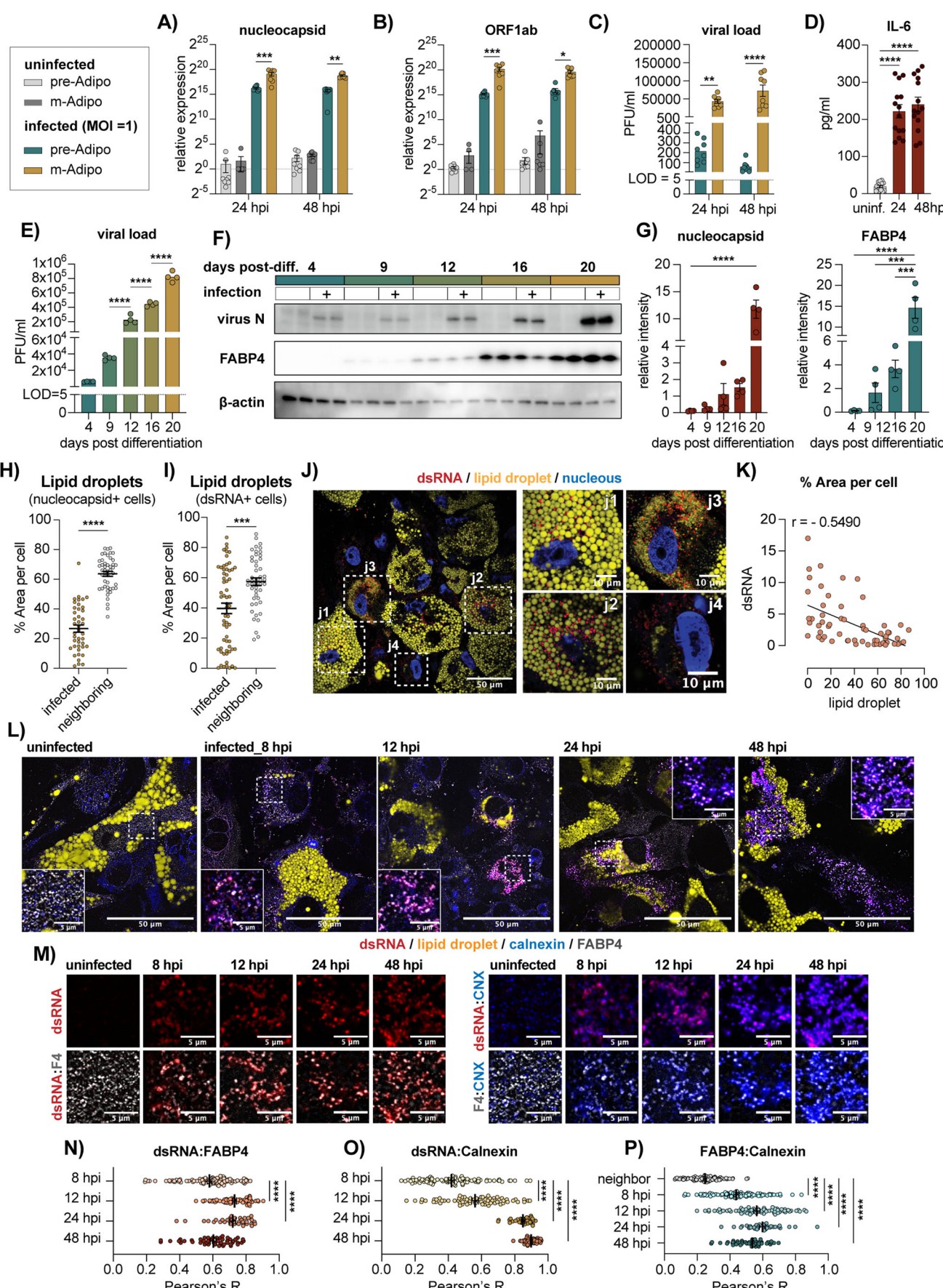

◀ **Figure 2. FABP4 colocalizes with SARS-CoV-2 replication organelles in human adipocyte cell lines.**

(A–C) hTERT pre-adipocytes and differentiated adipocytes infected with SARS-CoV-2 (WA1/2020, MOI = 1). (A, B) Relative expression of viral (A) genomic RNA (nucleocapsid, ***p = 0.0002, **p = 0.0018) and (B) sub-genomic RNA (ORF1ab, ***p = 0.0003, *p = 0.0263), normalized to β-actin. (C) Viral loads measured from supernatant using plaque assay. (****p < 0.0001, **p = 0.0024). Data are pooled from two independent experiments (n = 8, biological replicates). Statistical analysis was performed using two-way ANOVA. (D) IL-6 levels in the supernatant of differentiated adipocytes with or without viral infection (MOI = 1), measured by ELISA. Data are pooled from three independent experiments (n = 14, biological replicates, ****p < 0.0001). Statistical analysis was performed using one-way ANOVA. (E–G) Adipocytes were infected at 4, 9, 12, 16, and 20 days post-differentiation (MOI = 1), with measurements taken 48 h post-infection. (E) Viral loads measured by plaque assay; data represent two independent experiments (n = 4, biological replicates, ****p < 0.0001). Statistical analysis was performed using one-way ANOVA. (F) Western blot of viral nucleocapsid, FABP4, and β-actin proteins levels in cell lysates. (G) Quantification of nucleocapsid and FABP4 band intensities normalized to β-actin, representative of two independent experiments (n = 4, biological replicates, ****p < 0.0001, ***p = 0.0001 and 0.0006). Statistical analysis was performed using two-way ANOVA. (H, I) Percent area of lipid droplets in infected cells and neighboring cells, quantified by fluorescence neutral lipid staining (Bodipy). Infected cells identified by (H) nucleocapsid-positive staining and (I) dsRNA-positive staining. Data pooled from two independent experiments (n = 6, biological replicates, ****p < 0.0001, ***p = 0.0001); statistical analysis was performed using a standard t-test. (J) Representative confocal images of infected differentiated adipocytes (MOI = 1), stained for dsRNA (red), lipid droplets (yellow), and nucleus (DAPI, blue). Scale bar = 50 μm; magnified regions = 10 μm (n = 3). (K) Percent lipid droplet area relative to dsRNA-positive area per cell. Pearson correlation coefficient indicated as r. (L, M) Representative confocal images of control and infected adipocytes at 8, 12, 24, and 48 h post-infection, stained for dsRNA (red), lipid droplets (yellow), calnexin (blue), and FABP4 (gray). (L) Merged image of all stains (Scale bar = 50 μm); insets highlight overlap of FABP4, dsRNA, and calnexin (Scale bar = 5 μm). (M) Signal overlap between dsRNA, FABP4, and calnexin, and between FABP4 and calnexin individually (Scale bar = 5 μm). (N–P) Colocalization of target signals over time, represented as Pearson correlation R. Cells infected with MOI = 3 at 8 and 12 hpi; data pooled from two independent experiments (n = 6, biological replicates, ****p < 0.0001). For 24 and 48 hpi, MOI = 1 was used (n = 3) biological replicates. Statistical analysis was performed using one-way ANOVA. Data shown as mean ± s.e.m. Source data are available online for this figure.

ability to differentiate and accumulate LDs at levels comparable to control (Fig. EV3I). Infection of these cells confirmed that in the absence of FABP4, nucleocapsid protein abundance and viral titers were significantly reduced (Fig. 3F–H). To cross validate the specificity of the FABP4 inhibitors in reducing viral titers, we treated the control and FABP4$^{-/-}$ cells with increasing concentrations of either BMS309403 or CRE-14 and observed a dose-dependent reduction in virus nucleocapsid protein levels in the control but not the FABP4-deficient cells (Fig. EV3K–M). In addition, we generated a second human adipocyte cell line expressing FABP4-targeting shRNA to achieve long-term suppression of FABP4 and observed a similar reduction in viral infection (Fig. EV3J).

We then infected differentiated adipocytes with a higher viral dose (MOI = 3) to increase the incidence of dsRNA-positive cells, and thereby enable a better quantitative characterization of the dsRNA puncti. Interestingly, the dsRNA puncti in the FABP4-inhibitor treated samples appeared scattered across the cytoplasm failing to form coherent clusters (Fig. 3J), indicating a disruption of DMV formation in which they are packed. Similar results were observed in FABP4$^{-/-}$ adipocytes (Fig. 3K). A quantitative evaluation of the dsRNA-positive signal in the confocal images confirmed this observation, revealing a significant reduction in the percentage of dsRNA-positive area and mean fluorescence intensity in infected cells in the absence of FABP4 (Fig. 3I,L). We evaluated the LD content across control and inhibitor-treated cells and found that FABP4 inhibition did not affect LDs in uninfected cells or prevent their hydrolysis during infection (Fig. EV3N). We also infected the control and FABP4$^{-/-}$ cells at various stages of differentiation and found that while FABP4 deletion reduced the virus titers across all conditions, differentiated adipocytes still had higher viral titers than pre-adipocytes, even in the absence of FABP4 (Fig. EV3O,P). This indicates that LDs and FABP4 levels can independently promote viral replication. These observations in multiple chemical and genetic models demonstrated the critical importance of FABP4 for SARS-CoV-2 replication, likely through its engagement with the DMVs in the virus ROs.

We next asked whether the effect of FABP4 on viral infection also applies to other common cold coronavirus infections. We examined common cold coronavirus (HCoV-OC43) infection using a Real-Time Cell Electronic Sensing assay (RTCES) to measure cell viability and cytopathic effects (CPE) in wild type and FABP4-deficient mouse pre-adipocytes (Fig. 3M,N). We observed a significant reduction and delay in virus-induced CPE in the absence of FABP4. Similar results were also obtained in human lung fibroblasts (MRC5) following chemical inhibition of FABP4 (Fig. 3O). Taken together, these results indicate that the effect of FABP4 may be broader and apply to multiple coronavirus infections.

## FABP4 facilitates virus replication in bronchial epithelial cells

Having established FABP4's significance in SARS-CoV-2 adipocyte infection, we investigated its role in human bronchial epithelial (HBE) cells to understand whether it is engaged early on during respiratory tract infection (Beucher et al, 2022). HBE135-E6E7 cells were infected with high titers (MOI = 5) of various isolates of SARS-CoV-2 representing the alpha, delta, omicron, and Eris variants (Ank1, Ank-Dlt1, and Ank-Omicron GKS, Ank-Eris respectively). Following infection, cells were treated with two doses of both FABP4 inhibitors, and their viral titers were monitored over four days (Fig. 4A–D). In all tested conditions, FABP4 inhibition resulted in a marked reduction of virus titers examined over the time course of 96 h. Next, we used transmission electron microscopy (TEM) to examine the consequence of FABP4 inhibition on ROs morphology. Infected HBE cells (MOI:1) were treated with the FABP4 inhibitor (CRE-14) then fixed 48 h post-infection. TEM imaging revealed a marked reduction in the size of DMVs following FABP4 inhibitor treatment (Fig. EV4A,C).

We also utilized reconstructed airway epithelium organoids that were apically infected with SARS-CoV-2 and treated with 10 μM CRE-14 or 5 μM of the antiviral Remdesivir through the basal layer. Viral titers measured from the apical wash showed a reduction following inhibitor treatment 72 h post-infection that, at the higher

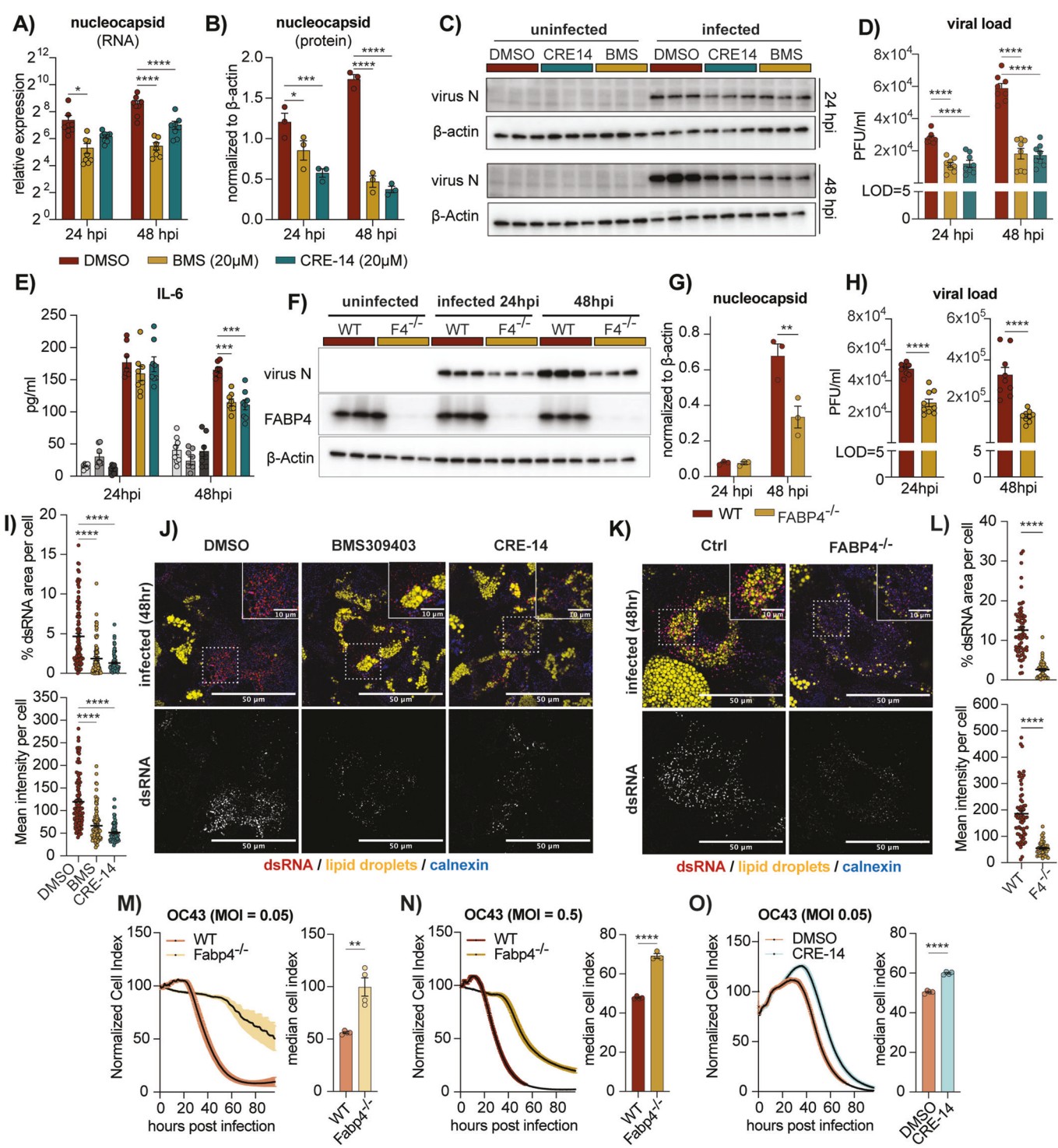

dose, matched the titers of the Remdesivir-treated group (Fig. 4E). We then examined the morphology and distribution of DMVs in the vehicle or CRE-14 treated organoids and observed that in the inhibitor-treated group DMVs were reduced in both size and numbers (Fig. 4F–I) and appeared more dispersed, failing to form the tight clusters observed in untreated controls (Fig. 4J,K). Clusters of LDs were observed in infected samples and were nearly absent in the inhibitor-treated group (Fig. 4L). Importantly, we

confirmed that exposure to FABP4 inhibitor in uninfected samples did not alter cellular or organelle morphology (Fig. 4B). We performed a similar analysis in samples treated with the BMS309403 compound, and though we observed similar results, the phenotype was less pronounced (Fig. EV4D–F), and the compound treatment showed signs of toxicity in these organoids. Overall, these data confirm the importance of FABP4 for SARS-CoV-2 replication through its influence on the ROs formation in

**Figure 3.  FABP4 deficiency reduces virus titers and disrupts replication organelles formation in adipocytes.**

**(A–E)** SARS-CoV-2-infected differentiated adipocytes (MOI = 1), treated with either DMSO, BMS309403 (20 μM) or CRE-14 (20 μM). **(A)** Relative RNA expression of SARS-CoV-2 nucleocapsid, normalized to β-actin. Data are pooled from two independent experiments ($n = 8$, biological replicates, ****$p < 0.0001$, *$p = 0.0229$). **(B)** Quantification of nucleocapsid band intensity normalized to β-actin. Data are representative of three independent experiments ($n = 3$, biological replicates, ****$p < 0.0001$, ***$p = 0.0003$, *$p = 0.0182$). **(C)** Western blot of nucleocapsid and β-actin protein levels in cell lysates. **(D)** Viral load measured by plaque assay, pooled from two independent experiments ($n = 8$, biological replicates, ****$p < 0.0001$), representative of four independent experiments. **(E)** IL-6 levels in the supernatant of infected adipocytes treated with FABP4 inhibitors, measured by ELISA. Data are pooled from two independent experiments ($n = 8$, biological replicates, ***$p = 0.0004$ and 0.0001). For **(A)**, **(B)**, **(D)**, and **(E)**, statistical analysis was performed using two-way ANOVA. **(F–H)** Infected wild-type (WT) and FABP4-deficient human adipocytes (MOI = 1). **(F)** Western blot of nucleocapsid, FABP4, and β-actin protein levels in cell lysates. **(G)** Quantification of nucleocapsid band intensity normalized to β-actin, representative of two independent experiments ($n = 3$, biological replicates, **$p = 0.0015$). **(H)** Viral load measured by plaque assay from supernatants collected 24 and 48 h post-infection. Data are pooled from three independent experiments ($n = 9$, biological replicates, ****$p < 0.0001$). Statistical analysis was performed using a standard t-test. **(I–L)** Infected differentiated adipocytes (MOI = 3, biological replicates, ****$p < 0.0001$) fixed 48 h post-infection and stained with dsRNA (red), lipid droplets (yellow), and calnexin (blue). **(J, K)** Representative images of infected cells with **(J)** inhibitor treatment or **(K)** genetic deletion of FABP4. Scale bar = 50 μm, magnified regions = 10 μm. **(I, L)** Percentage dsRNA area and mean fluorescence intensity per cell. Statistical analysis was performed using one-way ANOVA for **(I)** and a standard t-test for **(L)** ($n = 3$, biological replicates, ****$p < 0.0001$) biological replicates and 11 to 16 images per sample. **(M–O)** Real-time electric impedance traces and their corresponding median cell index (hours) for **(M)** and **(N)** wild-type and FABP4 knockout mouse pre-adipocytes infected with coronavirus OC43 at indicated MOIs. **(O)** MRC5 cells infected with OC43 and treated with either DMSO or CRE-14. Statistical analysis was performed using a standard t-test ($n = 3$, biological replicates, ****$p < 0.0001$, **$p = 0.0026$). Data are shown as mean ± s.e.m. Source data are available online for this figure.

multiple cellular targets. Its engagement with virus infection in airway epithelial cells and adipocytes, also increases its potential as a therapeutic target.

## FABP4 inhibition reduces viral titer and lung damage in infected hamsters

To examine the therapeutic potential of FABP4 inhibitor treatment in a preclinical model in vivo, we infected 12–14-week-old lean Syrian hamsters with SARS-CoV-2 (Ank1, 100 TCID$_{50}$) (Hanifeh-nezhad et al, 2020), and treated them for 6-days with a subcutaneous injection of 15 mg/kg FABP4 inhibitor BMS309403, CRE-14 or vehicle (Figs. 5 and EV5). The hamsters were monitored daily, and the experiments were terminated for virus titer measurements and histological analysis. The CRE-14 inhibitor treatment significantly ameliorated the infection-associated weight loss and reduced lung viral titers (Fig. 5A,B), while the BMS309403 treatment achieved similar but less pronounced results (Fig. EV5D,E). This might be explained in part by the higher plasma exposure and better pharmacokinetics of CRE-14 as assessed in mice (Fig. EV5A–C). The reduction in lung viral titers were also reflected in the immunofluorescence and immunohisto-chemical staining for the SARS-CoV-2 nucleocapsid protein in the vehicle and inhibitor-treated lungs (Fig. 5C–E).

We further examined the effect of FABP4 inhibition on lung pathology and observed a notable reduction in lung damage (Figs. 5F and EV5F). A careful evaluation of these samples performed by an independent pathologist showed a reduction in various clinical aspects associated with viral infections, including lung damage within bronchial and alveolar areas, as well as alveolar and subpleural edema (Table EV2 and Fig. 5G–L). Moreover, using the same sample set, we performed trichrome staining and observed increased collagen deposition in the lungs following infection which was reduced with inhibitor treatment (Figs. 5M and EV5G). Together, this data confirms that inhibition of FABP4 in vivo can reduce lung viral titers and alleviate lung damage and collagen deposition.

To get a sense of the effect of treatment on immune cell infiltration, and particularly macrophage infiltration, we performed CD68 immunohistochemical staining in the lungs of hamsters

infected with the Delta strain (Ank1-Dlt strain, 1000 TCID50). Using image analysis software, we then quantified the total cell number in each lung, as well as the number of CD68$^+$ cells, however, we found no significant changes across either parameter (Fig. EV5H–J).

## Discussion

In this study, we identify FABP4 as a critical host factor for SARS-CoV-2 infection and pathogenesis. We establish FABP4's importance in two separate human COVID-19 cohorts and demonstrate, through in vitro studies, that targeting FABP4 effectively reduces viral titers in multiple cell types and viral strains. In addition, the reduced cytopathic effects we observed in the absence of FABP4 in cells infected with the common cold coronavirus OC43, indicates that targeting FABP4 may be effective across various coronavirus infections. We also demonstrate that FABP4 inhibitor treatment of infected Syrian hamsters decreased viral loads and alleviated the infection-associated immunopathology.

Remodeling of the host intracellular membranes into virus replication organelles (ROs) is a strategy used by coronaviruses and other positive-stranded RNA (+RNA) viruses, to support their propagation and secretion and provide an environment shielded from immune recognition (Roingeard et al, 2022). In the case of coronaviruses, these compartments consist of DMVs, spherules, and zippering ER membranes. In this study, we observed a striking recruitment of FABP4 to ROs in infected cells. The FABP4 signal, which is usually diffused throughout the cytoplasm, condensed into puncti co-localized with dsRNA and calnexin signals, pointing to their recruitment to DMVs at the early stages of viral replication. Examination of infected adipocytes and bronchial epithelial cells with confocal microscopy and TEM revealed that the DMVs in cells where FABP4 was inhibited were smaller and more dispersed. We suggest that this loss of a spatial organization of the DMVs is what resulted in the reduced viral replication capacity, evident by the lower virus nucleocapsid RNA and protein, and overall viral titers. Investigating the exact role that FABP4 serves within the ROs is an exciting avenue to explore in future research, with potential implications for novel viral intervention strategies.

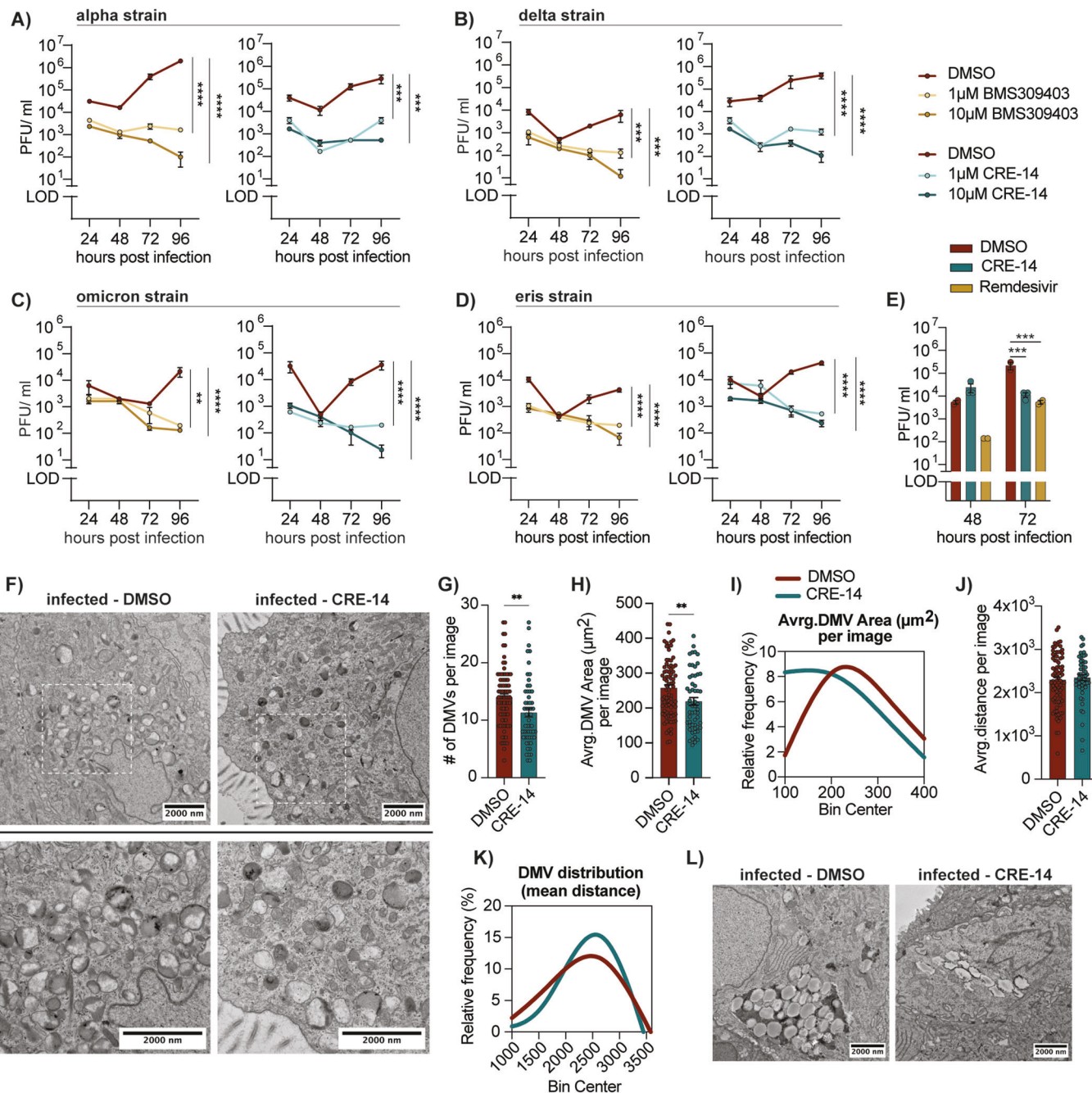

**Figure 4. Inhibition of FABP4 in bronchial epithelial cells disrupts virus replication organelles.**

(A–D) Viral load measured from the supernatant of human bronchoepithelial (HBE135-E6E7) cells infected with SARS-CoV-2 (MOI = 3, biological replicates) (A) alpha strain (Ank1, ****$p < 0.0001$, ***$p = 0.0003$), (B) delta strain (Ank-Dlt1, ****$p < 0.0001$, ***$p = 0.0002$ and 0.0001), (C) omicron strain (Ank-OmicronGKS), and (D) eris strain (Ank-Eris, ****$p < 0.0001$, ***$p = 0.0034$ and 0.0023) and treated with BMS309403 or CRE-14 at the indicated doses. Data are representative of two independent experiments. Statistical analysis was performed using two-way ANOVA. (E) Virus load measured from the apical wash of infected 3D airway epithelium cultures (IDF0571/2020, MOI 0.1), treated with either CRE-14 (10 μM) or Remdesivir (5 μM). Statistical analysis was performed using two-way ANOVA ($n = 3$, biological replicates, ***$p = 0.0002$ and 0.0004). (F–L) 3D reconstructed airway epithelium cultures were infected apically with $10^5$ PFU of SARS-CoV-2 (strain WA1/2020) then treated with either DMSO or CRE-14 through the basal layer. 24 h after infection, the cells were fixed for EM and 3 sections per sample ($n = 2$) were analyzed. (F, L) Representative TEM images showing (F) virus double-membrane vesicles (DMVs) and (L) lipid droplets in control (DMSO) and inhibitor treated samples. (G) Number of double membrane vehicles (**$p = 0.005$) and (H) their average area per image ($n = 2$, biological replicates, **$p = 0.0091$) with 18-45 images per sample taken at an 8000× magnification. (I) Frequency distribution of DMV area shown as a Fit Spline. (J) The mean distance between DMVs within each image, calculated from the X, Y coordinates of each DMV and (K) the percent frequency distribution shown as a fit spline. Statistical analyses were done using standard t-test. Data are shown as the mean ± s.e.m. Source data are available online for this figure.

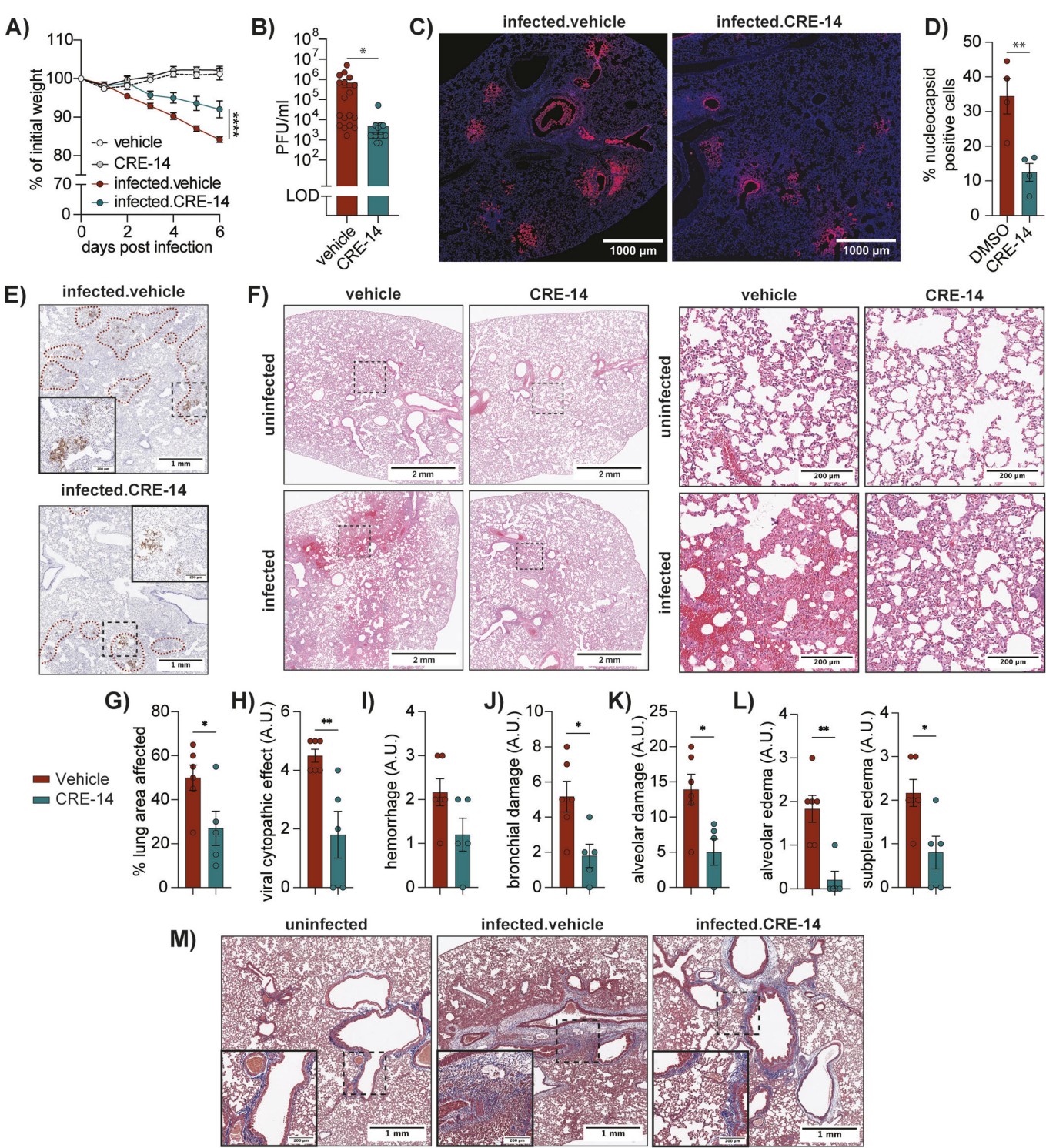

Previous studies have shown that SARS-CoV-2 and other viruses promote LD accumulation by upregulating de novo lipogenesis (DNL) while simultaneously increasing lipolysis within the first 24 h post-infection (Baek et al, 2022). Inhibiting components of either the DNL or lipolysis pathway, or disrupting LD contact sites with viral RO-associated ER membranes, significantly reduces viral titers (Williams et al, 2021; Ricciardi et al, 2022). In addition, cells transfected with viral non-structural proteins (nsp 3, 4, and 6) to form RO-like structures (ROLS) similarly accumulate and then consume LDs (Ricciardi et al, 2022). Consistent with these findings, and in contrast to the LD expansion seen in infected SGB-derived adipocytes (Quaranta et al, 2024), we observed a rapid depletion of LDs in infected adipocytes. This depletion was proportional to the density of their dsRNA puncta,

**Figure 5. FABP4 blockade limits virus replication and ameliorates pathology in infected hamsters.**

Syrian hamsters were infected intranasally with SARS-CoV-2 (Ank1, 100 TCID50) and treated daily with FABP4 inhibitor (CRE-14, 15 mg/kg) or vehicle. (A) Percent of initial body weight over time (B) Lung viral titers pooled from three independent experiments ($n = 18$ for infected vehicle, $n = 17$ for CRE-14 treated, and $n = 6$ for each uninfected group). Statistical analysis was performed using two-way ANOVA for (A) (****$p < 0.0001$) and a standard t-test for (B) (*$p = 0.028$). (C, E) Representative immunofluorescence and IHC staining of SARS-CoV-2 nucleocapsid in infected hamster lungs with or without CRE-14 treatment (Scale bar = 1 mm, $n = 4$). (D) Percentage of nucleocapsid-positive cells relative to total cell count, quantified from immunofluorescence staining ($n = 4$, biological replicates, **$p = 0.0086$). Statistical analysis was performed using a standard t-test. (F) Representative H&E staining of control and infected hamster lungs with or without inhibitor treatment (Scale bars: left = 2 mm, right = 200 μm). Low magnification images are shown in (Fig. EV5F). (G–L) Pathology evaluation of lung histology in arbitrary units (A.U.), based on pathology scores ($n = 6$ vehicle treated, $n = 5$ CRE-14 treated, further details in Table EV2). Statistical analysis was performed using a standard t-test ((G) *$p = 0.0391$, (H) **$p = 0.0063$, (J) *$p = 0.0158$, (K) *$p = 0.014$, (L) **$p = 0.002$ and *$p = 0.019$). (J) Bronchial damage represents combined pathology scores of bronchial epithelial cell necrosis and presence of cellular debris in bronchi. (L) Alveolar damage represents combined pathology scores of alveolar epithelial cell necrosis, cellular debris in alveoli, hyaline membranes, fibrin deposition, and alveolar emphysema. (M) Representative Masson's trichrome staining of control and infected hamster lungs with or without CRE-14 treatment (Scale bar = 1 mm, $n = 4$). Low magnification images are shown in (Fig. EV5G). Data are shown as mean ± s.e.m. Source data are available online for this figure.

suggesting that LDs are actively utilized during RO biogenesis. Although we expected FABP4 to play a role in promoting DNL during infection, its inhibition did not alter LD content or prevent infection-induced LD hydrolysis. Interestingly, our analysis of SARS-CoV-2 infection dynamics in human adipocytes revealed that differentiation significantly enhances viral replication capacity, even in the absence of FABP4. However, viral titers in FABP4$^{-/-}$ cells remained consistently and significantly lower than in the control cells. Therefore, we propose that the role of FABP4 in promoting viral replication lies downstream of lipolysis, potentially by facilitating lipid trafficking to viral RO membranes (Gericke et al, 1997; Herr et al, 1995; Wootan et al, 1993).

FABP4 has a marked impact on the metabolic dysregulation associated with obesity and aging (Charles et al, 2017). Its increase in obese mice and humans correlates positively with cardiometabolic pathologies and morbidity (Prentice et al, 2019). These correlations were clearly reflected in our patient cohorts and were associated with more severe COVID-19 disease. Furthermore, multiple prior GWAS studies reported that humans carrying a low-expression variant of FABP4 exhibit significant protection against the development of type 2 diabetes and cardiometabolic disease (Saksi et al, 2014; Zhao et al, 2017; Tuncman et al, 2006). Consequently, the impact of targeting FABP4 on the overall pathogenesis of COVID-19, and the protection that such interventions may offer to individuals with underlying metabolic conditions, may not be limited to the reduction in viral titers. In the context of obesity, we anticipate that the elevated levels of circulating FABP4, combined with increased lipid content in adipocytes, uncontrolled lipolysis, and ectopic lipid deposition in other organs (Maeda et al, 2005), may collectively contribute to the heightened disease severity.

FABP4 can also influence the inflammatory and metabolic milieu both within the local environment of the lungs, and systemically through its action as a secreted hormone. Increased FABP4 in pulmonary epithelial and endothelial cells drives immunopathology in various models of airway inflammation, and its inhibition or genetic deletion reduces proinflammatory cytokines, immune cell recruitment, and improves airway function (Shum et al, 2006; Ghelfi et al, 2013; Wang et al, 2017). Similarly, studies across various diseases, including atherosclerosis (Erbay et al, 2009), type 1 diabetes (Xiao et al, 2021), rheumatoid arthritis (Guo et al, 2022), and cancer (Yang et al, 2023), have identified FABP4 as a critical immune modulator that exacerbates disease pathology, primarily by enhancing macrophage proinflammatory

activity. FABP4 deletion can also affect CD8 T cell function, either indirectly through macrophage regulation (Xiao et al, 2021) or by directly modulating CD8 T cell lipid uptake (Pan et al, 2017). It is worth noting that, aside from adipocytes, endothelial cells and macrophages are two major sources of FABP4 (Prentice et al, 2019) and while neither cell type supports SARS-CoV-2 replication, their infection leads to increased proinflammatory cytokine production (Martínez-Colón et al, 2022; Wagner et al, 2021). Interestingly, FABP4$^+$ macrophage populations were significantly reduced in the bronchoalveolar lavage fluid of COVID-19 patients with severe disease compared to both controls and patients with moderate disease (Liao et al, 2020). Moreover, in our hamster studies, we observed no changes in macrophage infiltration in CD68$^+$ stained lung sections of infected inhibitor-treated hamsters. This, however, does not rule out a role for FABP4-macrophage regulation in promoting disease pathology, as the time of sampling as well as variations in inoculation titer and treatment dose could all influence these observations. It is also possible that the reduced lung damage and collagen deposition in inhibitor-treated, SARS-CoV-2-infected hamsters is a result of FABP4 inhibition in other cells, such as pulmonary endothelial and epithelial cells, or from suppressing FABP4's broader immunomodulatory activity across tissues and in circulation. Supporting this possibility, we observed a marked induction of IL-6 production in SARS-CoV-2-infected adipocytes, which was significantly diminished in the absence of FABP4. Given that IL-6 is a key clinical risk biomarker, its potential origin from adipose tissue presents an intriguing mechanism by which adipocyte infection could influence disease progression. Although the limitations of the hamster model have constrained our ability to fully investigate the effects of FABP4 inhibition or deletion on coronavirus disease pathophysiology, our findings provide a foundation for future studies to explore these mechanisms in greater depth.

We've shown in previous studies that antibody-mediated neutralization of FABP4 improves metabolic health in obese mice by lowering hepatic glucose production, improving insulin sensitivity, and reducing hepatic steatosis (Prentice et al, 2021; Burak et al, 2015). These effects may also counteract the COVID-19-associated hyperglycemia which is also attributed to increased hepatic glucose production and insulin resistance (Reiterer et al, 2021; Barreto et al, 2023). While this requires further investigation, a thorough examination of FABP4's engagement in these pathways could have important implications for mitigating the risk of developing diabetes or sustaining long-term respiratory symptoms,

both documented amongst the many symptoms of long-COVID (Davis et al, 2023; Xie and Al-Aly, 2022). Such investigations would benefit from the use of mouse models, for which we have a much more expansive toolkit of reagents, genetic, and pharmacological means of studying FABP4 biology, compared to hamster models. Thus, in future research, we hope to expand on the findings of this study using mouse-adapted coronavirus strains (Leist et al, 2020; Douglas et al, 2018).

FABP4 protein levels in the circulation are markedly elevated in humans with severe to critical clinical manifestations following SARS-CoV-2 infection, which points to the possibility of infected adipocytes producing higher levels of the protein. However, in the isolated cultured adipocytes, we did not detect an increase in FABP4 secretion upon infection. It is possible that adipocytes in their native tissue environment respond differently to infection, or that FABP4 secretion in vivo is elicited as a response to more complex systemic interactions, particularly those associated with the pathophysiology of severe COVID-19. This would explain why individuals with mild to moderate disease have significantly lower concentrations of FABP4. It is also possible that the source of FABP4 in circulation may be endothelial cells as they do provide a substantial proportion of circulating FABP4, especially at the baseline state (Inouye et al, 2023). This would be quite interesting to explore in the future in the cell type-specific FABP4-deletion mouse models.

In summary, our work demonstrates a critical contribution of FABP4 to SARS-CoV-2 infection as a host factor, and highlights FABP4 as a therapeutic target acting both as an antiviral and a modulator of cardio-pulmonary and metabolic fitness.

# Methods

**Reagents and tools table**

| Reagent/Resource | Reference or Source | Identifier or Catalog Number |
|---|---|---|
| **Experimental Models** | | |
| Patient samples | Koc University Medical School Hospital. | |
| hTERT human pre-adipocytes. | ATCC | CRL-3386 |
| Fabp4 knockout 3T3-L1 | In house | |
| Vero E6 cells | ATCC | CRL-1586 |
| HBE135-E6E7 | ATCC | CRL-2741 |
| MRC5 lung fibroblasts | ATCC | CCL-171 |
| HEK293 cells | ATCC | CRL-1573 |
| Airway epithelium organoids | MATTEK | AIR-100, AIR-112 |
| SARS-CoV-2 strain USA-WA1/2020 | BEI resources | NR-52281 |
| SARS-CoV-2 strain Ank1 | GenBank | MT478019 |
| SARS-CoV-2 strain Ank-Dlt1 | GenBank | OM295705 |
| SARS-CoV-2 strain Ank-omicron | GenBank | OR529199 |

| Reagent/Resource | Reference or Source | Identifier or Catalog Number |
|---|---|---|
| SARS-CoV-2 strain Ank-Eris | Partial sequencing | |
| SARS-CoV-2 strain SARS-CoV-2 BetaCoV | VirNext | IDF0571 |
| Human β-coronavirus OC43 | ATCC | VR-1558 |
| Syrian Hamsters (Mesocricetus auratus) | Janvier laboratorie | RjHan:AURA |
| **Recombinant DNA** | | |
| lentiCRISPRv2 | Addgene | 52961 |
| **Antibodies** | | |
| Rabbit anti-SARS-CoV-2 nucleocapsid | Abcam | ab271180 |
| Mouse anti-SARS-CoV-2 nucleocapsid | Cell Signaling Technology | 33717 |
| Rabbit anti-β-actin | Abcam | ab8224 |
| Rabbit anti-GAPDH | Cell Signaling Technology | 5174S |
| Mouse anti-FABP4 | Dana Fabor Antibody Core | 351.4.2E12.H1.F12 |
| Mouse anti-FABP4 | Dana Fabor Antibody Core | 351.4.5E1.H3 |
| Rabbit anti-FABP4 | Abcam | ab216708 |
| Rabbit anti-FABP4 | Abcam | 13979 |
| Goat anti-FABP4 | Novus | AF1443 |
| Rabbit anti-FABP4 | Sigma | HPA002188 |
| Mour anti-double stranded RNA J2 | Exalpha | 10010500 |
| Rabbit anti-calnexin | Cell Signaling Technology | 2679 |
| Anti-mouse secondary antibody | Cell Signaling Technology | 4410 |
| Anti-rabbit secondary antibody | ThermoFisher Scientific | A-11037 |
| Anti-rabbit HRP-conjugated antibody | Abcam | ab64264 |
| Anti-SARS-CoV-2 nucleocapsid | GeneTex | GTX635686 |
| **Oligonucleotides and other sequence-based reagents** | | |
| FABP4 targeting shRNA | Origene | TL313105 |
| Scrambled shRNA | Origene | TR30021 |
| SARS-CoV-2 nucleocapsid | ThermoFisher Scientific | Vi07918637_s1 |
| SARS-CoV2 ORF1ab | ThermoFisher Scientific | Vi07921935_s1 |
| Human FABP4 | ThermoFisher Scientific | Hs01086177_m1 |
| Human β-actin | ThermoFisher Scientific | Hs01060665_g1 |
| **Chemicals, Enzymes and other reagents** | | |
| DMEM/high glucose media | ThermoFisher Scientific | 11965118 |
| FBS | Atlanta Biologicals | S11550 |
| Penicillin streptomycin | Lonza | 17-603E |
| Human insulin | Sigma-Aldrich | I9278 |
| Biotin | Sigma-Aldrich | B4639 |
| Panthothenate | Sigma-Aldrich | P-5155 |

| Reagent/Resource | Reference or Source | Identifier or Catalog Number |
|---|---|---|
| Dexamethasone | Sigma-Aldrich | D-1756 |
| Triiodothyronin | Sigma-Aldrich | T-6397 |
| IBMX | Sigma-Aldrich | I-5879 |
| Indomethacin | Sigma-Aldrich | I8280 |
| BCS | Hycline | 16777-206 |
| Rosiglitazone | Cyman | 71740 |
| HEPES | ThermoFisher Scientific | 15630080 |
| TEER buffer | MATTEK | TEER-BUFFER |
| CRE-14 | Crescenta Biosciences | Patent 17/566692 |
| BMS309403 | MedKoo | 524464 |
| Remdesivir | MedChemExpress | HY-104077 |
| Bovine serum albumin | Carl Roth | 9048-46-8 |
| HPMC | Merck | H7509 |
| Tween-80 | Sigma-Aldrich | P1754 |
| Methylcellulose | Sigma-Aldrich | M0512 |
| Formaldehyde | Fisher Scientific | BP531-500 |
| Crystal violet | Sigma-Aldrich | C0775-25G |
| Puromycin | Sigma-Aldrich | P4512 |
| Qiazol reagent | Qiagen | 79306 |
| RIPA buffer | ThermoFisher Scientific | 89900 |
| Orthovanadate | New England Biolabs | P0758L |
| Inhibitor cocktail | Sigma-Aldrich | P8340 |
| Pierce 660nm Protein Assay Reagent | ThermoFisher Scientific | 22660 |
| 4x Laemmle buffer | Bio-Rad | 1610747 |
| β-mercaptoethanol | ThermoFisher Scientific | 21985023 |
| TGX Stain-Free Protein gels | BioRad | 5678095 |
| Blotting grade blocker | BioRad | 170-6404 |
| VXL buffer | Qiagen | 1069974 |
| Paraformaldehyde (PFA) | Electron Microscopy Sciences | 15710-S |
| TritonX100 | Sigma-Aldrich | 9002-93-1 |
| BodipyTM 493/503 | ThermoFisher Scientific | D3922 |
| Mounting media containing Dapi | Vector Laboratories | H-1800-10 |
| Glutaraldehyde | Electron Microscopy Sciences | ..... |
| **Software** | | |
| ImageJ | | |
| QuPath (0.5.0) | | |
| GraphPad Prism (10.1.1) | | |
| **Other** | | |
| NucleoSpin RNA isolation kit | Mecherey-Nagel | 740955 |
| iScript cDNA synthesis kit | BioRab | c1708897 |
| StepOne RT-qPCR kit | NEM Luna | E3005 |

| Reagent/Resource | Reference or Source | Identifier or Catalog Number |
|---|---|---|
| ONE-GloTM Luciferase assay kit | Promega | E6110 |
| IL-6 Quantikine ELISA kit | R&D Systems | D6050 |
| RTCES Multiplate System | Acea | .... |
| 1.5 coverslip | MatTek | P35GCOL-1.5-14-C |

## Cell lines

Human Telomerase Reverse Transcriptase (hTERT) pre-adipocytes (ATCC, CRL-3386) (Lee et al, 2004), were maintained in DMEM/high glucose media (ThermoFisher Scientific, 11965118) with 10% FBS (Atlanta Biologicals, S11550), 1% penicillin streptomycin (Lonza, 17-603E). Cells were differentiated 2 days after reaching confluence using DMEM/high glucose with 2% FBS, 1% penicillin streptomycin, human insulin (0.5 μM, Sigma-Aldrich, I9278), biotin (33 μM, Sigma-Aldrich, B4639), panthothenate (17 μM, Sigma-Aldrich, P-5155), dexamethasone (0.1 μM, Sigma-Aldrich, D-1756), Triiodothyronin (2 nM, Sigma-Aldrich, T-6397), IBMX (500 μM, Sigma-Aldrich, I-5879), Indomethacin (30 μM, Sigma-Aldrich, I8280). Cells were maintained in differentiation media for 18–20 days prior to the start of infection.

Wild type and *Fabp4* knockout 3T3-L1 mouse adipocyte cell lines (Furuhashi et al, 2007) were maintained in DMEM/high glucose 10% BCS (Hycline, 16777-206), and 1% penicillin streptomycin, then differentiated two days after reaching confluence with DMEM/high glucose, 10% FBS, 1% penicillin streptomycin, 5 mg/ml insulin, 500 mM IBMX, and 2 mM rosiglitazone (Cyman, 71740). Cells were kept in differentiation media for 2 days then switched to growth media with 1 mg/ml insulin.

Vero-E6 cells (ATCC, CRL-1586) were cultured in DMEM/high glucose with 10%FBS, 1% penicillin streptomycin, 1% HEPES (ThermoFisher Scientific, 15630080). HBE135-E6E7 human bronchoepithelial cells (ATCC, CRL-2741), were cultured in DMEM/high glucose with 10%FBS, 1% penicillin streptomycin. Human MRC5 lung fibroblasts (ATCC, CCL-171) were cultured in DMEM/high glucose with 10% FBS, 4 mM L-glutamine, and 1% penicillin streptomycin. Airway epithelium organoids (MATTEK, AIR-100, AIR-112) were cultured following manufacturer's instructions. Viral titers measurements in airway epithelium organoids were performed by Epithelix Sarl, France (www.epithelix.com), using MucilAir™ human 3D tissue from airways and lung surgical pieces, following manufacturer's instruction. FBS used in all cellular experiments was heat inactivated. Cell lines and periodically monitored for Mycoplasma contamination. No STR profiling is done.

## Viruses

SARS-CoV-2 (USA-WA1/2020) isolate seed stock was acquired from BEI resources (NR-52281) and propagated in-house in Vero-E6 cells. SARS-CoV-2 (Ank1) was isolated from nasopharyngeal swabs of patients with COVID-19-related respiratory illness in January 2020, Ankara, Turkey, and shown to be an alpha variant of SARS-CoV-2 (B.1.1.7) (Hanifehnezhad et al, 2020), (GenBank Acc.

No: MT478019). The SARS-CoV-2 Delta, Omicron and Eris variants were isolated from patient nasopharyngeal swabs in September 2021, April 2022, and May 2023 respectively. [Ank-Dlt1 (B.1.617.2) (GenBank Acc. No: OM295705), and Ank-Omicron GKS (BA.1.17) (GenBank Acc. No: OR529199)]. The variant identification was done after genome-wide sequence analysis following two-step plaque purification for the Delta and Omicron variants. The Eris variant was identified through partial sequencing of the spike coding region. SARS-CoV-2 BetaCoV/France/IDF0571/2020 strain was used in the MucilAir™ 3D human airway epithelial cell infections.

Human β-coronavirus OC43 (ATCC, VR-1558), was propagated in MRC5 cells. All infections were performed using growth media (corresponding to the infected cell) with 2% heat inactivated FBS. Target cells were incubated with a low volume of virus-containing media for 1 h, with gentle shaking every 15 min. After infection, cells were washed with PBS before adding fresh growth media. 3D airway epithelium cultures were infected through the apical layer after it was washed with TEER buffer (MATTEK, TEER-BUFFER, PBS with $Mg^{2+}$ and $Ca^{2+}$). The apical layer was washed again following infection, and liquid was removed. All in vitro experiments involving SARS-CoV-2 were performed in Biosafety level 3 facilities at the Department of Immunology and Infectious Diseases at Harvard T.H Chan School of Public Health and Faculty of Veterinary Medicine, Ankara University, following the approval from the Harvard Environmental Health and Safety, and the local Ethical Committee for Animal Experiments at Ankara University. VirNext provided commercial services performed their in-house facilities (France, https://virnext.fr).

## Human data

Ethics approval was obtained from Koç University Institutional Review Board (IRB) for Clinical Research (Approval number: 2020.136.IRB1.026). Lung biopsies were acquired between June–September of 2020 from a control subject and 3 COVID-19 patients: patient 1 (age 61) underwent a biopsy for squamous cell carcinoma, and patients 2 and 3 (ages 46 and 64) underwent biopsy for hemoptysis symptoms. All patients tested positive for COIVD-19 on the day of biopsy. The control lung biopsy was acquired from an 85-year-old male patient with squamous cell carcinoma, collect in 2021, with a negative COVID-19 test. Serum samples from COVID-19 patients and healthy controls were collected at the Koç University Medical School Hospital. Informed consent was obtained from all human subjects and the experiments conformed to the principles set out in the WMA Declaration of Helsinki and the Department of Health and Human Services Belmont Report.

## Hamster infection

Specific pathogen free (SPF) Syrian Hamsters (Mesocricetus auratus, 12–14 weeks old, males and females of 120 g average body weight at time of infection) were purchased from Janvier laboratories, France (name GOLDHAMSTER, strain RjHan:AURA), and then maintained in the SPF animal facility at the Biotechnology Institute, Ankara University, accredited by the Ministry of Agriculture and Forestry, Turkey. Hamsters were housed under a 12:12 light: dark cycle and fed ad libitum. All in vivo SARS-CoV-2 studies were conducted in Animal Biosafety

level 3 laboratories in the Faculty of Veterinary Medicine, Ankara University and with the approval of the Local Ethical Committee for Animal Experiments, Ankara University (06/May/2020, 2020-08-66 and/09/2021, 2021-16-150). Humane endpoint scores were considered and multiple observations per day were conducted to confirm the animals' welfare. To minimize bias hamsters were randomly assigned to different experimental groups. No blinding was performed during data collection. Hamsters were anesthetized using a combination of 100 mg/kg Ketamine and 7 mg/kg xylasine, then infected intranasally with SARS-CoV-2 (Ank1) dissolved in serum-free DMEM. Inhibitor treatments were administered daily by subcutaneous injection, starting simultaneously with infection. At the experiment endpoint (on day 6 post-infection), hamsters were euthanized after anesthesia. All animal samplings were conducted according to the national regulations on the operation and procedure of animal experiment ethics committees (Regulation Nr: 26220, Date: 09.7.2006).

## FABP4 inhibitors

CRE-14 was manufactured and provided by Crescenta Biosciences Inc. as described in the US Patent 17/566692 published on July 6, 202 (Koyuncu et al, 2023). The CRE-14 interaction with FABP4 was evaluated using a TR-FRET-based ligand displacement assays and micro-scale thermophoresis assay (MST). The ligand displacement assay was performed by Cepter Biopartners, New Jersey, (https://www.cepterbiopartners.com/), using a Terbium (Tb)-based time-resolved fluorescence energy transfer (TR-FRET) assay. Briefly, CRE-14 and BODIPY FL C12 were prepared at a concentration of 1.085 mM and 4.2 µM, respectively, in DMSO. 1.2 µL of each compound or DMSO (vehicle control) and 1.2 µL of BODIPY FL C12 were added into the wells of a 384-well black polypropylene plate. His6-FABP4 and Tb anti-His6 antibody were prepared in the assay buffer (25 mM Tris/HCl, pH 7.4, 0.4 mg/ml γ-globulins, 0.010% NP-40, 1 mM DTT) at a concentration of 83 nM and 49.6 nM, respectively. The protein and antibody solutions were then mixed at a ratio of 34:7 (v/v) and incubated on ice for 30 min. The assay was initiated by adding 41 µL of the resulting protein/antibody solution into the wells containing the compounds and BODIPY FL C12. The plate was centrifuged and incubated at room temperature for 10 min. The TR-FRET signals were detected using an EnVision Multilabel plate reader (PerkinElmer; TB excitation 320 nm, BODIPY FL C12 emission 520 nm; TB emission 615 nm). Relative fluorescence ratio (520 nm × 10,000/615 nm) was used to calculate the compound mediated inhibition of BODIPY C12 FL fatty acid binding to FABP4. The same procedure was performed with the BMS309403 compound. The MST assay was performed by 2bind GmbH, Germany (https://2bind.com/). Recombinant human FABP4 produced E. coli was labeled with a fluorescent dye NT-650-NHS 2nd generation fluorescent dye (Nanotemper) in the labeling buffer (1 x PBS pH 7.4, 1 mM TCEP, 2.5% DMSO) containing 10 µM of the recombinant FABP4 and 30 µM of the dye at 25 °C. Following a 30-min incubation the solution was passed through PD Minitrap™ G-25 columns to eliminate the excess dye. Serial dilutions of CRE-14 were prepared in the assay buffer and mixed with the labeled FABP4 in a final volume of 10 µl of the assay buffer (1 x PBS pH 7.4, 1 mM TCEP, 0.05% Pluronic F-127, 2% DMSO). The solution was filled into premium-coated glass capillaries (NantoTemper) and analyzed on a Monolith NT.115 Pico machine

(red-pico; NanoTemper) at 25 °C. The final concentration of the labeled FABP4 was 10 nM and the CRE-14 concentration ranged between 20 μM to 610 pM (15 × 2-fold dilutions of the highest concentration of the compound). The binding affinity constants ($K_D$) for FABP4 were estimated by fitting the dose-response curve using non-linear regression using GraphPad Prism software.

BMS309403 (MedKoo, 524464) is a highly selective FABP4 inhibitor. Further information on this compound was previously reported (Sulsky et al, 2007). For cellular experiments FABP4 inhibitors (BMS309403, CRE-14) were dissolved in sterile DMSO (Cell Signaling Technology, cat. no. 12611S), to create a stock solution of 10 mM, which were aliquoted and stored at −20 °C. Remdesivir (MedChemExpress, HY-104077) was diluted in DMSO and used at 5 μM. The impact of CRE-14 on cell viability was assessed by incubating MRC5 cells with increasing concentrations of the compound using media containing either 2% or 10% FBS. 72 h after incubation, cell viability was measured using ONE-Glo™ Luciferase assay kit (Promega, E6110). Prior to cell treatments, a solution of the desired concentration of the inhibitors were created by dissolving the inhibitors in the respective cell growth mediums, containing 2% heat inactivated FBS or 1 mg/ml bovine serum albumin (Carl Roth, 9048-46-8). Inhibitor treatments were initiated 1-hour after cell incubation with infection media.

For in vivo treatment, inhibitors were dissolved in 0.5% hydroxypropyl methylcellulose (HPMC) (Merck, H7509) containing 1% Tween-80 (Sigma-Aldrich, P1754) and the pH of the drug was adjusted to 8.0 before injection. A solution of (0.5% HPMC and 1%T-80, pH 8.0) was used as vehicle. The Pharmacokinetics of BMS309403 and CRE-14 in vivo were assessed in C56BL/6J mice and Syrian hamsters following subcutaneous injection. Blood samples were collected at the indicated intervals and the compounds' concentration in the plasma were assessed by LC MS/MS.

## Virus infectivity assays

For plaque assay quantification of viral loads, ~$2 \times 10^5$ cells/well of Vero-E6 cells were seeded in 12-well plates 24 h prior to infection. 10-fold serial dilutions of the supernatant samples were created using DMEM/high glucose media with 2% heat inactivated FBS and 1%HEPES. After removing the Vero cell growth media, 200 μl of samples were added and incubated for 1 h with gentle shaking every 15 min. After incubation, an overlay of (1:1) growth media and 2% methylcellulose (Sigma-Aldrich, M0512) was added, and the cells were incubated at 37 °C for ~72 h. The overlay was then removed and 4% formaldehyde (Fisher Scientific, BP531-500) in PBS was used to fix the cells for 20 min. Cells were then washed with PBS, and then stained using crystal violet solution to quantify plaques [2%(w/v) crystal violet (Sigma-Aldrich, C0775-25G), with 20% methanol in distilled water]. Viral titers measured from HBE cells, 3D reconstructed airway organoids and hamster lung viral titers were evaluated as TCID50 measurements (50% tissue culture infectious dose), then converted to plaque forming units (PFU = TCID50 × 0.7). Here cytopathic effects were continually observed under a light microscope for each time point, and the TCID50 was calculated using the Reed and Muench method. Lung viral loads in the hamster studies were measured from the inferior and post-caval lobes of the lungs after homogenization in DMEM.

## CRISPR deletion and shRNA targeting FABP4

Genetic deletion of *FABP4* in human adipocytes cells (hTERT) was performed using lentiviral CRISPR plasmids. The lentiviral CRISPR plasmids were generated by cloning single guide RNA targeting FABP4 or control sgRNA into lentiviral CRISPR backbone lentiCRISPRv2 (Addgene, 52961). HEK293 cells (ATCC, CRL-1573] were transfected with the lentiviral plasmids using Lipofectamine™ LTX Reagent with PLUS™ Reagent (Thermo-Fisher Scientific, 15338100). Lentivirus particles were collected from the filtered supernatant and used to infect hTERT cells (at a 40–50% confluence) with the addition of 5 μg/ml polybrene (EMD Millipore, TR-1003-G). hTERT cells were re-plated from 6 cm to 10 cm dishes once they reached 80% confluence and were subjected to 1 μg/ml puromycin (Sigma-Aldrich, P4512) selection.

hTERT cell lines stably expressing shRNA targeting FABP4 or a scrambled control were generated using lentiviral shRNA plasmids (Origene, TL313105, and TR30021, respectively). HEK293 cells were transfected using Lipofectamine™ LTX Reagent with PLUS™ Reagent, and lentiviral particles were collected from filtered supernatant and used to infected hTERT cells with the addition of 8 μg/ml Polybrene. After 24 h, virus medium was removed, and a fresh growth media was added. After another 24 h, hTERT cells were re-plated onto 10 cm dishes and subjected to 3 μg/ml puromycin selection. To validate the CRISPR knockout and shRNA knockdown of FABP4, cells were differentiated and FABP4 protein levels were examined at different stages of differentiation.

3T3-L1 mouse adipocytes lacking *Fapb4* were reported earlier by our laboratory (Furuhashi et al, 2007), and were maintained and differentiated using the same protocol as the wild-type 3T3-L1 adipocytes.

## RNA isolation and qPCR

Qiazol reagent (Qiagen, 79306) was used to inactivate SARS-CoV-2 and collect RNA from cell lysates following the manufacturer's instructions. The RNA-containing aqueous phase was then added to an equal volume of 70% ethanol and RNA isolation was conducted using the NucleoSpin RNA isolation kit (Mecherey-Nagel, 740955). cDNA synthesis was conducted using the iScript cDNA synthesis kit (BioRab, c1708897), and the qPCR was performed using Taqman reagents [ThermoFisher Scientific; SARS-CoV-2 *nucleocapsid* (Vi07918637_s1), SARS-CoV-2 *ORF1ab* (Vi07921935_s1), human *FABP4* (Hs01086177_m1), human *β-actin* (Hs01060665_g1)].

In hamster studies, virus RNA was extracted from lung homogenate of the superior right lobe in 1 ml of Triazole Reagent (ThermoFisher Scientific, 15596026). Virus genomic RNA for the spike encoding gene was measured using the previously described primer/probe set (SF1, SR1, and SPr) (Hanifehnezhad et al, 2020). Amplifications were performed using a StepOne RT-qPCR kit (NEM Luna, E3005). Standard curves were generated using plasmids encoding the relevant regions of SARS-CoV-2, and the threshold for detection of fluorescence above the background was set within the exponential phase of the amplification curves. CT values from each sample were converted into log10 viral RNA copies/mg tissue according to the standard curve.

## Western blot analysis

We used RIPA buffer (ThermoFisher Scientific, 89900), supplemented with orthovanadate (New England Biolabs, P0758L) and protease inhibitor cocktail (Sigma-Aldrich, P8340). Protein concentrations were determined using Pierce 660 nm Protein Assay Reagent (ThermoFisher Scientific, 22660). Lysates were diluted in 4x Laemmle buffer (Bio-Rad, 1610747) and β-mercaptoethanol (ThermoFisher Scientific, 21985023), then boiled for 5 min at 95 °C. Samples were subjected to gel electrophoresis in 4–20% Criterion TGX Stain-Free Protein gels (BioRad, 5678095), before being transferred onto PVDF or nitrocellulose membranes using BioRad Transblot Turbr semi-dry transfer system. Gels were then blocked using Blotting grade blocker (Bio-Rad, 170-6404). The following antibodies were used in this study: SARS-CoV-2 nucleocapsid (Abcam, ab271180—1:1000 dilution), β-actin (Abcam, ab8224—1:1000 dilution), GAPDH (Cell signaling technology, 5174S—1:1000 dilution). FABP4 was detected using an in-house antibody produced for the Hotamisligil laboratory by the Dana Farber Antibody Core (HRP-tagged clone 351.4.5E1.H3—1:1000 dilution). This FABP4 antibody was validated using with protein lysates from FABP4-knockout mice as negative controls and recombinant FABP4 as positive control.

## ELISA

Supernatant of infected cells were mixed with VXL buffer (Qiagen, 1069974) in a 1:1 ratio, which has been shown to be sufficient to inactivate SARS-CoV-2 (Pastorino et al, 2020). FABP4 secretion was measured by in-house FABP4 ELISA using anti-FABP4 antibodies produced for the Hotamisligil laboratory by the Dana Farber Antibody Core (clone 351.4.2E12.H1.F12 for capture, and HRP-tagged clone 351.4.5E1.H3 for detection—1:1000 dilution) and recombinant human FABP4 (R&D Systems, DY3150-05) as a standard. IL-6 was measured following manufacturer's instructions using the human IL-6 Quantikine ELISA kit (R&D Systems, D6050).

## Real-time cellular electronic sensing (RTCES) assay

96-well E-Plates (Acea) were blanked with high glucose DMEM (Gibco) supplemented with 10% FBS (Gibco), 4 mM L-Glutamine (ThermoFisher), and 10 U/mL Penicillin Streptomycin (Gibco) to create a baseline reading before seeding the cells at a density of $2.6 \times 10^4$ cells per well. Plates were incubated at room temperature in a biosafety cabinet for 1 h before being placed into the RTCES Multiplate System (Acea) for 24 h within a 37 °C incubator. Cell impedance was recorded every 15 min for every active well. Media was removed and virus or mock in 2% heat-inactivated FBS, 4 mM L-Glutamine, and 10 U/mL Penicillin Streptomycin was added to the cells at a multiplicity of infection (MOI) of 5, 0.5, and 0.05. Plates were returned to the RTCES Multiplate System and incubated at 35 °C. RTCES measurements were taken every 15 min for 4 days. The median cell index values were determined by fitting the normalized cell index values to a sigmoidal curve using GraphPad Prism software 9.3.1.

## Confocal imaging

Cells were grown and differentiated in collagen-coated glass bottom 35 mm imaging dishes, with a no.1.5 coverslip (MatTek, P35GCOL-

1.5-14-C). At the indicated time points cells were fixed for 20 min using 4% paraformaldehyde (PFA; Electron Microscopy Sciences, 15710-S) in equal volumes of PBS and growth media. After fixation cells were kept in PBS at 4 °C until staining. Prior to staining cells were permeabilized with 0.2% TritonX100 (Sigma-Aldrich, 9002-93-1) in PBS for 20 min on a slow shaker at room temperature. Cells were then washed twice with PBS. Primary antibodies were diluted in PBS and added to the cells overnight at 4 °C. After removing the antibodies, cells were washed two to three times with PBS then incubated with the secondary antibody (diluted 1:1000 in PBS) for 1 h at room temperature. Cells were washed again prior to lipid droplet chemical staining with Bodipy™ 493/503 (Thermo-Fisher Scientific, D3922) diluted 1:1000 in PBS. After another wash, the cells were mounted in mounting media containing Dapi (Vector Laboratories, H-1800-10) was added to the cells. The following antibodies were used for this study: anti-SARS-CoV-2 nucleocapsid mouse antibody (Cell Signaling Technology, 33717—1:200 dilution), anti-double stranded RNA J2 mouse antibody (Exalpha, 10010500—1:200 dilution), anti-calnexin rabbit antibody (Cell Signaling Technology, 2679—1:200 dilution), anti-FABP4 rabbit monoclonal antibody (Abcam, ab216708—1:200 dilution), anti-FABP4 goat polyclonal antibody (Novus, AF1443—1:200 dilution), anti-mouse secondary antibody (Cell Signaling Technology, 4410—1:1000 dilution), anti-rabbit secondary antibody (ThermoFisher Scientific, A-11037—1:1000 dilution).

Image analysis was performed using ImageJ. To analyze the signal per cell, cells were manually selected and added to the ROI manager by tracing the visible borders of the stained components. After splitting the channels, the background was subtracted, and the intensity histogram of each ROI was analyzed. A set minimum threshold was then defined for each measured signal, and the % area-limited to threshold for each cell was measured. Co-localization analysis was performed using the ImageJ plug-in JACop.

## Transmission electron microscopy (TEM) imaging

HBE cells were suspended using 0.25% trypsin in EDTA, then fixed with 4% PFA in equal volumes PBS and growth media. Cells were incubated for 20 min at room temperature, then centrifuged at 6000 rpm at 4 °C for 10 min. The pellet was then resuspended in a 1:1 mix of growth media and an EM fixative buffer and incubated at 4 °C until TEM processing. The EM fixative buffer contained: [1.25% formaldehyde, 2.5% glutaraldehyde, 0.03% picric acid, 0.05 M cocadylate buffer]. For 3D reconstructed airway epithelium organoids, the 4% PFA and fixative buffer were added directly to the apical and basal layers. Samples were washed several times in 0.1 M cocadylate buffer, then post fixed with 1% osmiumtetroxide $(OsO_4)$/1.5% potassium ferrocyanide $(K_2FeCN_6)$ for 3 h followed by several washings in water. 1% uranyl acetate in maleate buffer was added for 1 h, then washed several times in maleate buffer (pH 5.2). The samples were then dehydrated in graded cold ethanol series up to 100% (50%, 70%, 90% then 100%, 10 min each) followed by 1 h incubation in propylene oxide. After the 70% ethanol dehydration step, 3D airway epithelium cultures were cut out of the insert with a scalpel. Samples were then incubated overnight at 4 °C in a 1:1 mixture of propylene oxide and TAAB 812 Resin (TAAB Laboratories Equipment). The following day, samples were embedded in TAAB resin and polymerized at 65 °C

for 48 h. 80 nm sections were cut with a Leica Ultracut S microtome and placed on formyar-carbon coated slot Cu grids, then stained with 0.2% Lead Citrate and viewed using a JEOL 1200EX electron microscope at 80 kV.

Image analysis was performed using ImageJ software on images acquired at a set magnification and the scale bar was used to determine the pixel/micron ratio. The DMVs were manually selected and added to the ROI manager and the DMV area and X, Y centroid coordinates were measured. To determine the DMV distribution, the distances between the DMV centroids were measured based on their X-Y coordinates and the (mean, median, minimum, and maximum) distances per image were calculated.

### Histological imaging

Human lung biopsies were provided by Koc University Department of Pathology and Yedikule Hospital, Department of Pathology, Istanbul Turkey. After Paraffin embedding and sectioning the samples were deparaffinized under 60 °C for 30 min then underwent a series of washes with PBS with 0.05% Tween 20. The slides were then incubated for 10 min in hydrogen peroxide. Antigen retrieval was performed by manually boiling with 1X citrate (pH 6.0) for 10 min. Protein blocking was done for 10 min. The cells were then incubated at room temperature with anti-FABP4 antibody (Sigma, HPA002188) at a 1:400 dilution in PBS. Secondary antibody was done using anti-rabbit HRP-conjugated antibody (Abcam, ab64264). Sections were then counterstained with hematoxylin.

For hamster histology analysis, the left lobe of the lung from each hamster was fixed in 4% PFA for 48 h prior to transfer out of the BSL3 facility. H/E staining and IHC were done by the Histowiz automated histology platform (https://home.histowiz.com/). The following antibodies were used for the IHC staining anti-FABP4 antibody (Abcam, 13979), anti-SARS-CoV-2 nucleocapsid antibody (GeneTex, GTX635686). Cell quantification was performed using QuPath (0.5.0). Briefly, we measured the total tissue area per lung and used the built-in cell detection function to quantify the total cell number per lung. We then identified nucleocapsid[+] cells using a machine learning classifier trained on manually selected nucleocapsid-positive and nucleocapsid-negative cells. A similar approach was used to quantify the total cell number and percentage of CD68[+] cells.

### Statistical analysis

For the analysis of human cohort 1, the COVID-19 patient data was combined with the healthy controls, making a total of 326 individuals and 1040 observations, with an average cluster size of 3.2. The main association analyses were performed using the generalized linear mixed model (LMM), regressing the FABP4 concentration on the COVID-19 severity (the reference group consists of the healthy individuals), while accounting for the time of collection post symptom onset, as well as patient-level covariates: age, sex, and BMI. Because the estimates from LMM can be biased under model mis-specification, an alternative modeling framework for longitudinal analyses was applied—the Generalized Estimation Equations (GEE)—using the exchangeable correlation structure. Since GEE is robust to incorrectly specified correlation structures

**The paper explained**

**Problem**

Metabolic disorders such as obesity and aging increase the risk and severity of respiratory infections like COVID-19. Fatty acid-binding protein 4 (FABP4), a key immunometabolic regulator, has been implicated in promoting obesity- and age-related pathologies. In this study, we investigated the role of FABP4 as a host factor in SARS-CoV-2 infection and evaluate its potential as a therapeutic target.

**Results**

We observed elevated FABP4 levels in the lungs and circulation of COVID-19 patients, which strongly correlated with disease severity. In cellular systems, we found that FABP4 is recruited to viral replication organelles, and its inhibition or deletion significantly reduced viral replication across multiple cell types and SARS-CoV-2 variants. In hamsters, FABP4 inhibition led to reduced lung viral titers and mitigated lung pathology.

**Impact**

Our study identifies FABP4 as a critical host factor in coronavirus replication and lung pathology. We demonstrate that targeting FABP4 presents a promising therapeutic approach to reduce viral replication and improve outcomes, particularly in patients with metabolic comorbidities. These findings open new avenues for antiviral strategies that could enhance infection control and metabolic health, while minimizing the risk of emerging resistance.

within the clusters, we performed sensitivity analyses by comparing the estimates from GEE to those based on the LMM. We observed similar results, which lends support to the usage of LMM (i.e., indicating the absence of model mis-specification). As a secondary analysis, we replaced COVID-19 severity with oxygen support measures, as a categorical variable, and investigated the association between FABP4 and oxygen support, conditional on the time of collection post symptom onset, age, sex, and obesity as covariates. We used the same LMM model as previously described. We combined the individuals who required nasal canula and mask into one group and treated those with no need for oxygen support as the reference group.

Other statistical analyses were performed using GraphPad Prism version 10.1.1. for MacOS. GraphPad Software, San Diego, California, USA, www.graphpad.com. All data are presented as mean ± s.e.m.

## Data availability

The source data of this paper are collected in the following database record: biostudies:S-SCDT-10_1038-S44321-024-00188-x.

## Peer review information

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

## Acknowledgements

We thank all members of the Hotamisligil Laboratory, past and present, for their scientific support and input during the development of this project. We also thank Drs. Phyllis Kanki, and Don Hamel for their support in facilitating our work in the BSL-3 facility at Harvard Chan School, Dr. Lynn Enquist of Princeton University for his valuable feedback, Dr. Pinar Firat from Koç University and Dr. Nur Urer from the Ministry of Health Yedikule State Hospital for help with human histological samples. Electron microscopy image analysis was conducted with assistance from the Image Analysis Collaboratory at Harvard Medical School (https://iac.hms.harvard.edu/). Support by Hotamisligil Lab and Crescenta Biosciences for their part of the experiments.

## Author contributions

**Hatoon Baazim**: Conceptualization; Data curation; Formal analysis; Validation; Investigation; Visualization; Writing—original draft; Writing—review and editing. **Emre Koyuncu**: Formal analysis; Supervision. **Gürol Tuncman**: Project administration. **M Furkan Burak**: Project administration. **Lea Merkel**: Formal analysis; Investigation. **Nadine Bahour**: Investigation; Performed Sample Analysis. **Ezgi Simay Karabulut**: Investigation. **Grace Yankun Lee**: Investigation; Methodology. **Alireza Hanifehnezhad**: Investigation. **Zehra Firat Karagoz**: Investigation. **Katalin Földes**: Investigation. **Ilayda Engin**: Investigation. **Ayse Gokce Erman**: Investigation. **Sidika Oztop**: Investigation; performed histological sample preparations and analyzed the data. **Nazlican Filazi**: Investigation. **Buket Gul**: Investigation. **Ahmet Ceylan**: Investigation. **Ozge Ozgenc Cinar**: Investigation. **Fusun Can**: Data curation; Investigation. **Hahn Kim**: Validation; Investigation; Methodology. **Ali Al-Hakeem**: Visualization. **Hui Li**: Data curation. **Fatih Semerci**: Visualization. **Xihong Lin**: Data curation. **Erkan Yilmaz**: Data curation. **Onder Ergonul**: Supervision; Investigation. **Aykut Ozkul**: Supervision; Investigation. **Gökhan S Hotamisligil**: Conceptualization; Supervision; Funding acquisition; Writing—review and editing.

In addition to the CRediT author contributions listed above, the contributions in detail are:

HB designed and performed the in vitro and cell-based experiments and analyzed the data, performed the histology, fluorescence, and electron microscopy image analysis, prepared the figures, and wrote and revised the manuscript. EK designed, supervised, and performed experiments relating to the CRE-14 compound and OC43 virus infection, interpreted results, revised the manuscript, and contributed to the conception of the project. GT and MFB provided intellectual contributions and coordinated collaborations. LM generated the human FABP4 knockout cell lines and performed sample analysis. NB and ESK performed sample analysis. GYL provided intellectual insight and assistance in generating the human FABP4 knockout cell lines. AH, ZFK, and KF performed in vivo hamster experiments. IE, AGE, and EY maintained the animal colonies and designed and prepared animal cohorts and supervised animal experiments. SO, AC, OOC, and FS performed histological sample preparations and analyzed the data. NF and BG performed in vitro assays and analyzed data. FC and OE conducted all human studies and performed the FABP4 measurements in human samples. HK designed and synthesized the CRE-14 compound and provided intellectual contributions and contributed to the conception of the project. AA-H established the RTCES assay protocols and analyzed data. HL and XL performed the statistical analysis for the human data. AO planned, supervised, and conducted all hamster infection experiments and analysis of the results and revised the manuscript. GSH conceived, supervised and supported the project, designed experiments, interpreted results and planned and revised the manuscript.

Source data underlying figure panels in this paper may have individual authorship assigned. Where available, figure panel/source data authorship is listed in the following database record: biostudies:S-SCDT-10_1038-S44321-024-00188-x.

## Disclosure and competing interests statement

Emre Koyuncu is co-founder, director, and officer of Crescenta Biosciences and holds equity at the company. Ali Al-Hakeem and Fatih Semerci are employees of Crescenta Biosciences. Hahn Kim is co-founder, director, and consultant of Crescenta Biosciences and holds equity at the company. Hahn Kim is an employee of Princeton University; All work of Hahn Kim included herein was performed as a consultant for Crescenta, independent of Princeton University. Hahn Kim, Emre Koyuncu and Gökhan S. Hotamisligil are inventors on patent application that includes CRE-14. Gökhan S. Hotamisligil is a scientific advisor, receives compensation and holds equity at Crescenta Biosciences. He is also a member of the journal's advisory editorial board. This has no bearing on the editorial consideration of this article for publication. Other authors declare no competing interests.

# Expanded View Figures

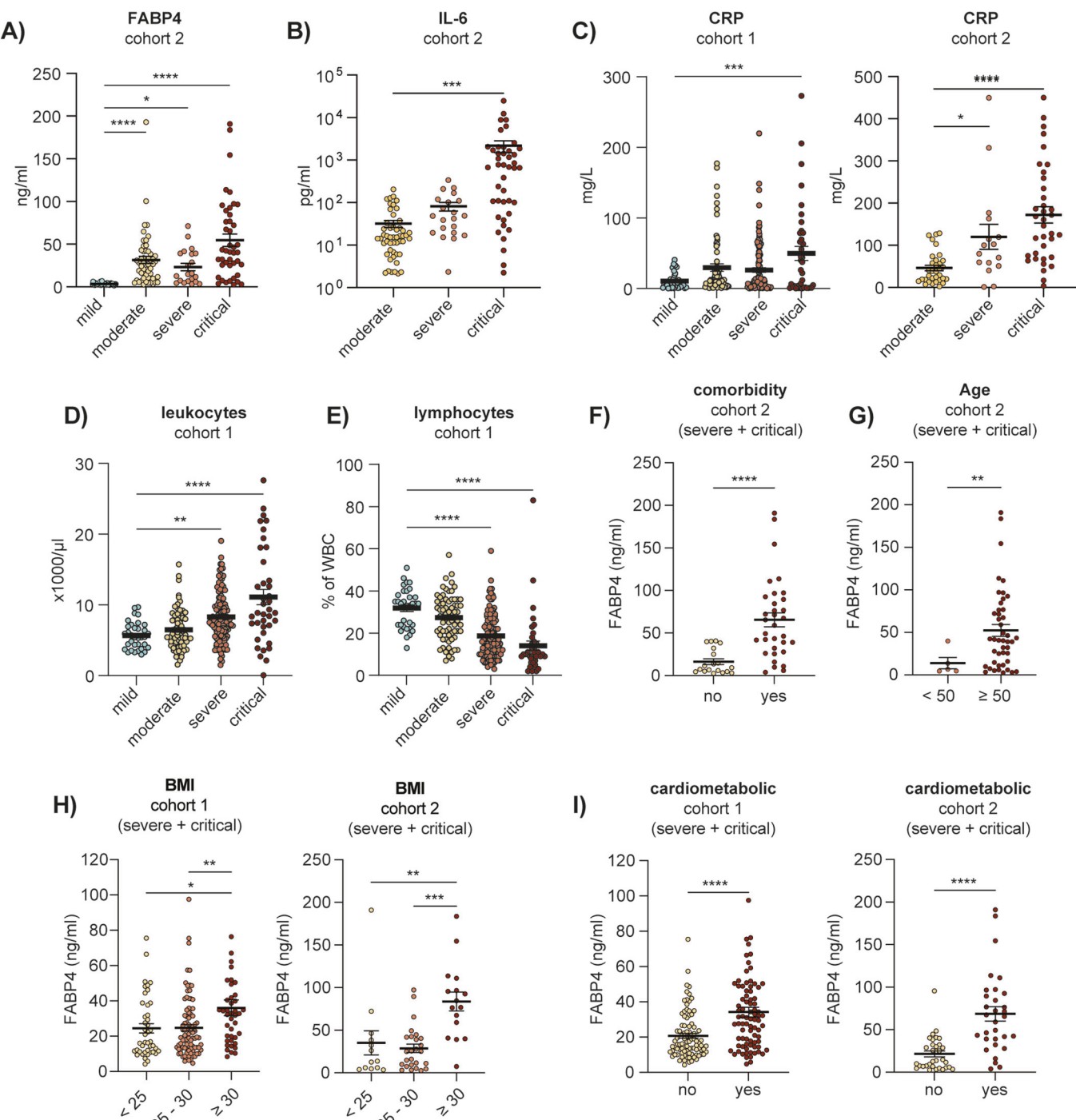

**Figure EV1.  Increase in FABP4 along with biomarkers of COVID-19 disease severity.**

(A, B) Maximum concentrations of circulating (A) FABP4 (****$p < 0.0001$, *$p = 0.044$) and (B) IL-6 (***$p = 0.0005$) of cohort 2 of COVID-19 patients ($n = 166$) stratified based on disease severity (moderate: $n = 52$, severe: $n = 21$, and critical: $n = 42$). (C) Circulating levels of C-reactive protein (****$p < 0.0001$, ***$p = 0.0002$, *$p = 0.0221$), (D) leukocytes (****$p < 0.0001$, **$p = 0.0015$) and (E) lymphocytes (****$p < 0.0001$) of COVID-19 patients measured on the day in which the maximum FABP4 concentration was measured (day post symptom onset). Statistical analysis was performed using one-way ANOVA ($n = 283$ cohort 1, and $n = 116$ cohort 2). (F–I) Maximum FABP4 concentration pooled from severe and critically ill patients (cohort 1: $n = 176$, cohort 2: $n = 63$), stratified based on (F) the presence or absence of comorbidities (listed in Table 3 and Dataset EV2, ****$p < 0.0001$), (G) age (**$p = 0.0011$), (H) BMI (***$p = 0.0001$, **$p = 0.0098$ and 0.004, *$p$ 0.0263), and (I) the presence or absence of cardiometabolic conditions (diabetes, hypertension, or coronary artery disease, ****$p < 0.0001$). Statistical analysis for (F), (G) and (I) were performed using Welch's t-test and one-way ANOVA for (H). Data are shown as the mean ± s.e.m. Source data are available online for this figure.

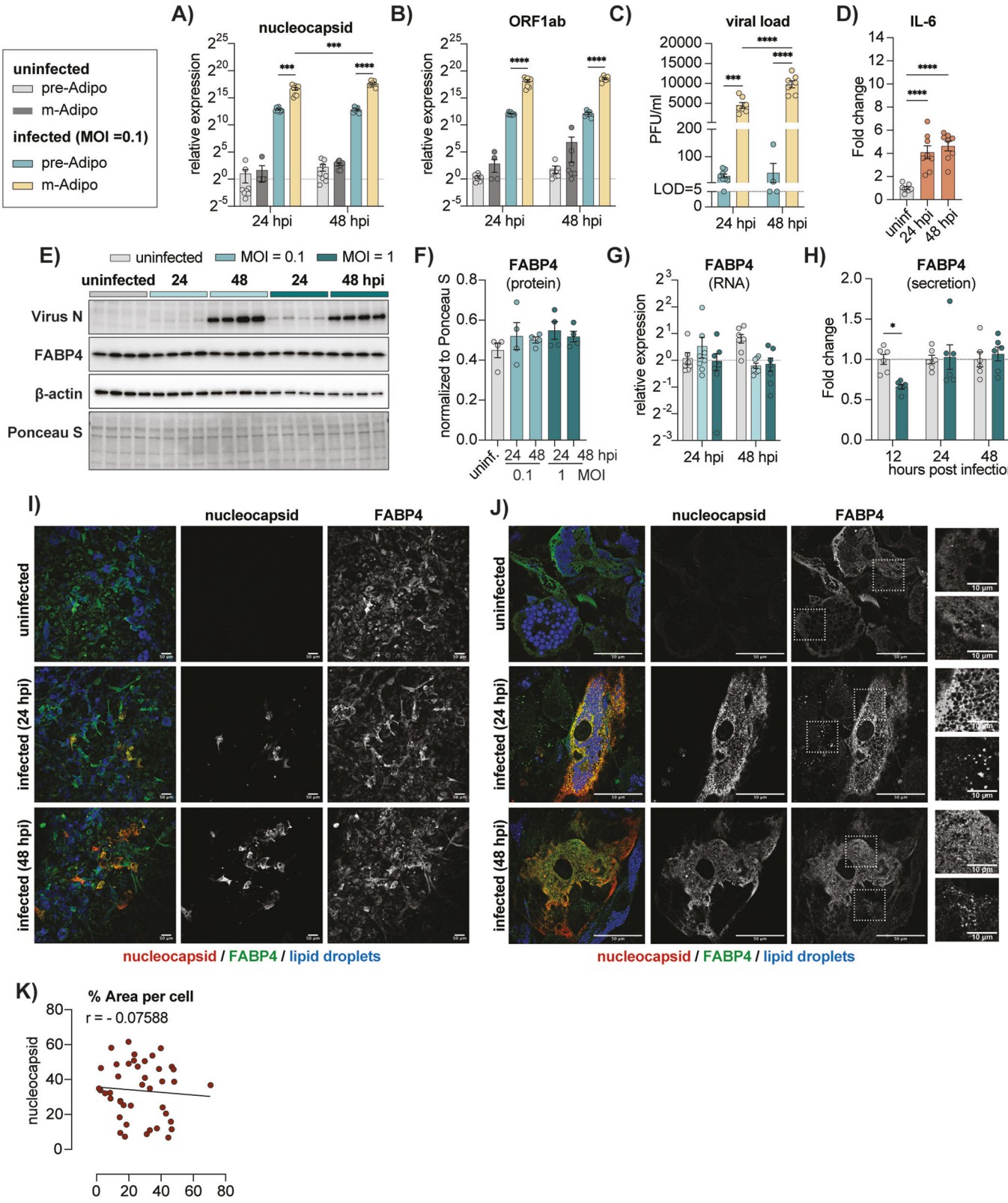

nucleocapsid / FABP4 / lipid droplets

◀ **Figure EV2.  FABP4 regulation during SARS-CoV-2 infection.**

(A–D) Pre-adipocytes and differentiated adipocytes infected with SARS-CoV-2 (WA1/2020, MOI = 0.1). (A, B) Relative expression of viral (A) genomic RNA (nucleocapsid, ****$p < 0.0001$, ***$p = 0.0003$ and 0.0004) and (B) sub-genomic RNA (ORF1ab, ****$p < 0.0001$), normalized to β-actin. (C) Viral loads measured from supernatant using plaque assay (****$p < 0.0001$,***$p = 0.0001$). Data are pooled from two independent experiments ($n = 8$, biological replicates). Statistical analysis was performed using two-way ANOVA. (D) IL-6 measured by ELISA in the supernatant of differentiated adipocytes with or without virus infection (MOI = 0.1). Data are pooled from two independent experiments ($n = 8$, biological replicates, ****$p < 0.0001$). Statistical analysis was performed using one-way ANOVA. (E) Western blots of SARS-CoV-2 nucleocapsid, FABP4, β-actin protein levels, and total protein (Ponceau S staining) in cell lysates of differentiated adipocytes infected with SARS-CoV-2 (MOI = 0.1 or MOI = 1). (F) Quantification of FABP4 band intensity normalized to total protein, representative of two independent experiments ($n = 4$, biological replicates). (G) FABP4 gene expression relative to β-actin, pooled from two independent experiments ($n = 8$, biological replicates). (H) FABP4 secretion in the supernatant within 1-hour incubation at the indicated time points following infection. Fold change is calculated relative to uninfected samples. Data are representative of two independent experiments ($n = 6$, biological replicates, *$p = 0.0336$). Statistical analysis was performed using two-way ANOVA. (I, J) Representative confocal images of infected adipocytes stained with nucleocapsid (red), FABP4 (green), and lipid droplets (blue). (I) Low magnification and (J) high magnification images of the same samples (Scale bars = 50 μm, magnified regions = 10 μm) ($n = 3$, biological replicates). (K) Percentage lipid droplet area relative to nucleocapsid-positive area per cell in infected differentiated adipocytes. Pearson correlation coefficient is indicated as r. Data are shown as the mean ± s.e.m. Source data are available online for this figure.

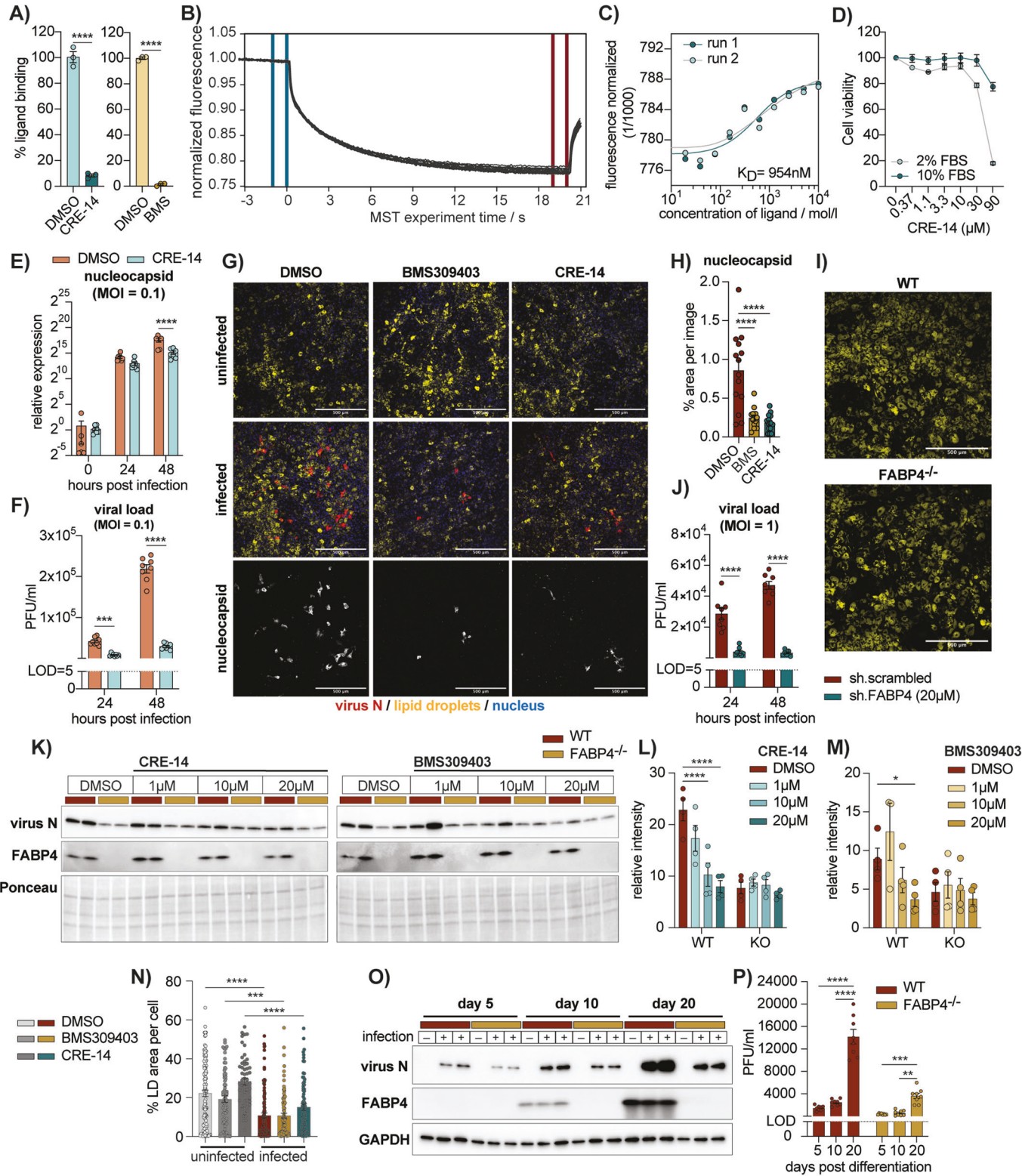

◀ **Figure EV3. FABP4 deficiency reduces virus titers and cell death following coronavirus infection.**

(A) Percentage of FABP4 bound with fatty acid (BODIPY FL C12) in the presence or absence of CRE-14 or BMS309403 ($n = 3$, technical replicates, ****$p < 0.0001$). (B) Representative MST time traces with blue and red regions indicating $F_{cold}$ and $F_{hot}$, respectively, from which fluorescence measurements were normalized. (C) Dose-response curve showing FABP4 binding to increasing concentrations of CRE-14, represented as normalized fluorescence. KD value (954 nM) represents the average across two technical runs. (D) MRC5 cell viability following administration of titrated doses of CRE-14 at the indicated concentrations of FBS. (E, F) Differentiated adipocytes infected with SARS-CoV-2 (WA1/2020, MOI = 0.1) and treated with either CRE-14 (20 μM) or DMSO. (E) Relative RNA expression of nucleocapsid normalized to β-actin (****$p < 0.0001$). (F) Viral load measured from the supernatant using plaque assay (****$p < 0.0001$, ***$p = 0.0004$). (G) Representative confocal images of control and infected adipocytes (MOI = 1), fixed 48 h post-infection, stained for virus nucleocapsid (red), lipid droplets (yellow), and nuclei (DAPI, blue) ($n = 3$, biological replicates). Scale bar = 500 μm. (H) Percentage of nucleocapsid-positive area per image, averaging 4–5 images per sample (****$p < 0.0001$). (I) Representative confocal images showing lipid droplet content (yellow) in WT and FABP4-deficient adipocytes. (J) *FABP4*-shRNA knockdown and scrambled controls infected with SARS-CoV-2 (WA1/2020, MOI = 1), with viral titers measured by plaque assay from supernatants. Data are pooled from two independent experiments (n = 8, biological replicates, ****$p < 0.0001$). Statistical analysis was performed using two-way ANOVA. (K–M) WT and FABP4-deficient differentiated adipocytes infected and treated with either DMSO, CRE-14, or BMS309403 at indicated doses, with cell lysates collected 48 h post-infection. Data are representative of two independent experiments ($n = 3$). (K) Western blots showing nucleocapsid, FABP4, and total proteins (Ponceau S staining) in cell lysates. (L, M) Quantifications of nucleocapsid band intensity relative to total protein. Statistical analysis was performed using two-way ANOVA (****$p < 0.0001$, *$p = 0.017$). (N) Percentage of lipid droplet area per cell in uninfected and SARS-CoV-2-infected adipocytes with or without FABP4 inhibitor treatment (20 μM, ****$p < 0.0001$, ***$p = 0.0006$). (O, P) Adipocytes infected at 5, 10, and 20 days post-differentiation (MOI = 1). (O) Western blot of nucleocapsid, FABP4, and GAPDH protein levels in cell lysates, and (P) viral titers in supernatant measured 48 h post-infection. Data are representative of two independent experiments ($n = 3$, biological replicates, ****$p < 0.0001$, ***$p = 0.0008$, **$p = 0.0016$). Statistical analysis was performed using two-way ANOVA. Data are shown as mean ± s.e.m. Source data are available online for this figure.

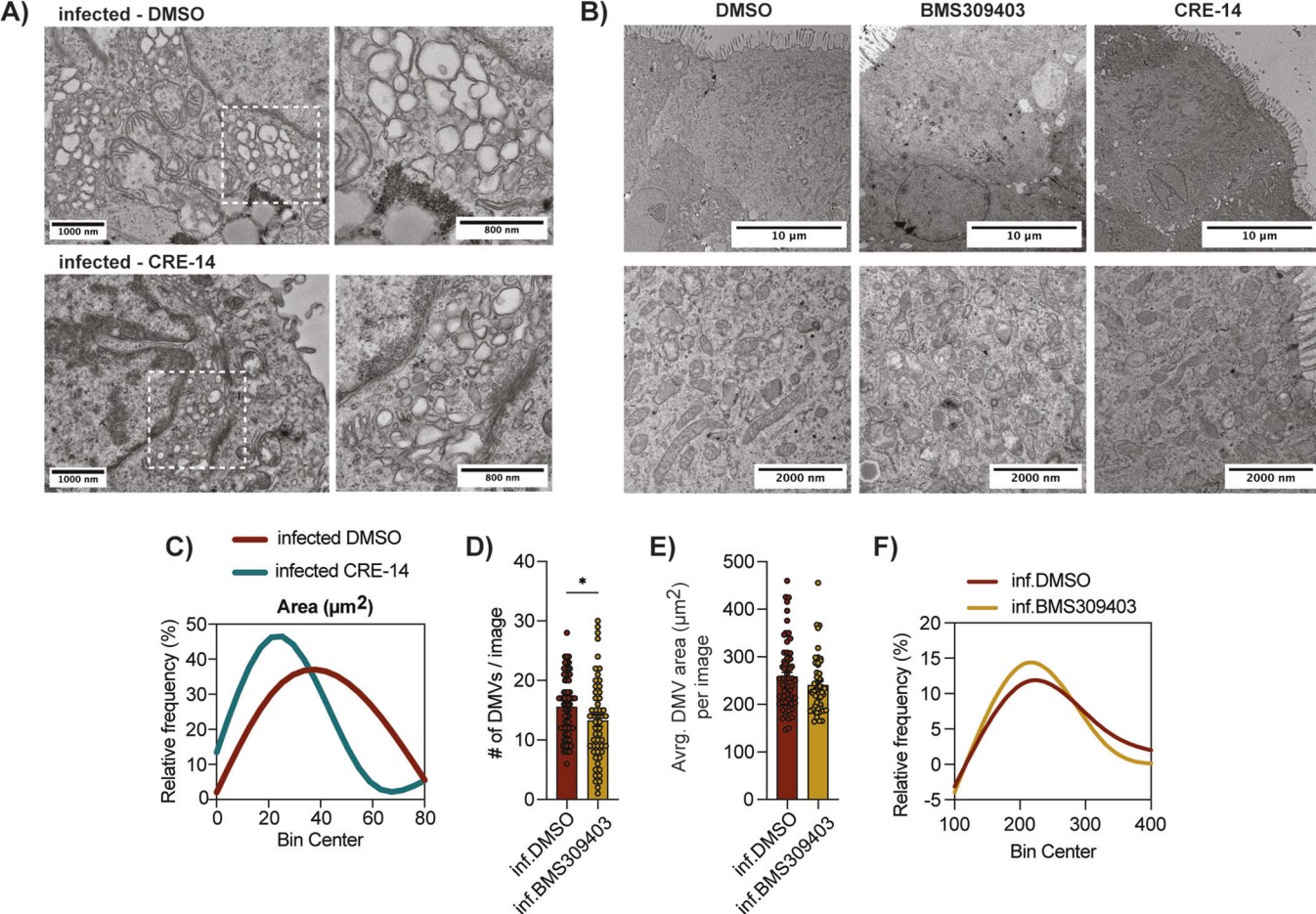

**Figure EV4. FABP4 inhibition reduces viral titers across various SARS-CoV-2 variants.**

(**A, B**) Representative transmission electron microscopy (TEM) images of (**A**) HBE cells 48 h after infection with SARS-CoV-2 (WA1/2020, MOI = 1) and treatment with DMSO or CRE-14 (10 μM) ($n = 3$, biological replicates). (**B**) Uninfected reconstructed airway epithelium 3D culture treated with DMSO, BMS309403, or CRE-14. (**C**) Area of double membrane vesicles in infected HBE cells, determined from TEM images in (**A**). Data displayed as the Fit Spline of the percent frequency distribution. (**D**) Number of DMVs per image (*$p = 0.0242$), (**E**) average DMV area per image, and (**F**) its frequency distribution quantified from TEM images reconstructed airway epithelium cultures infected with SARS-CoV-2 and treated with DMSO or BMS309403. Statistical analysis is performed using standard t-test ($n = 3$, biological replicates). Data are shown as the mean ± s.e.m. Source data are available online for this figure.

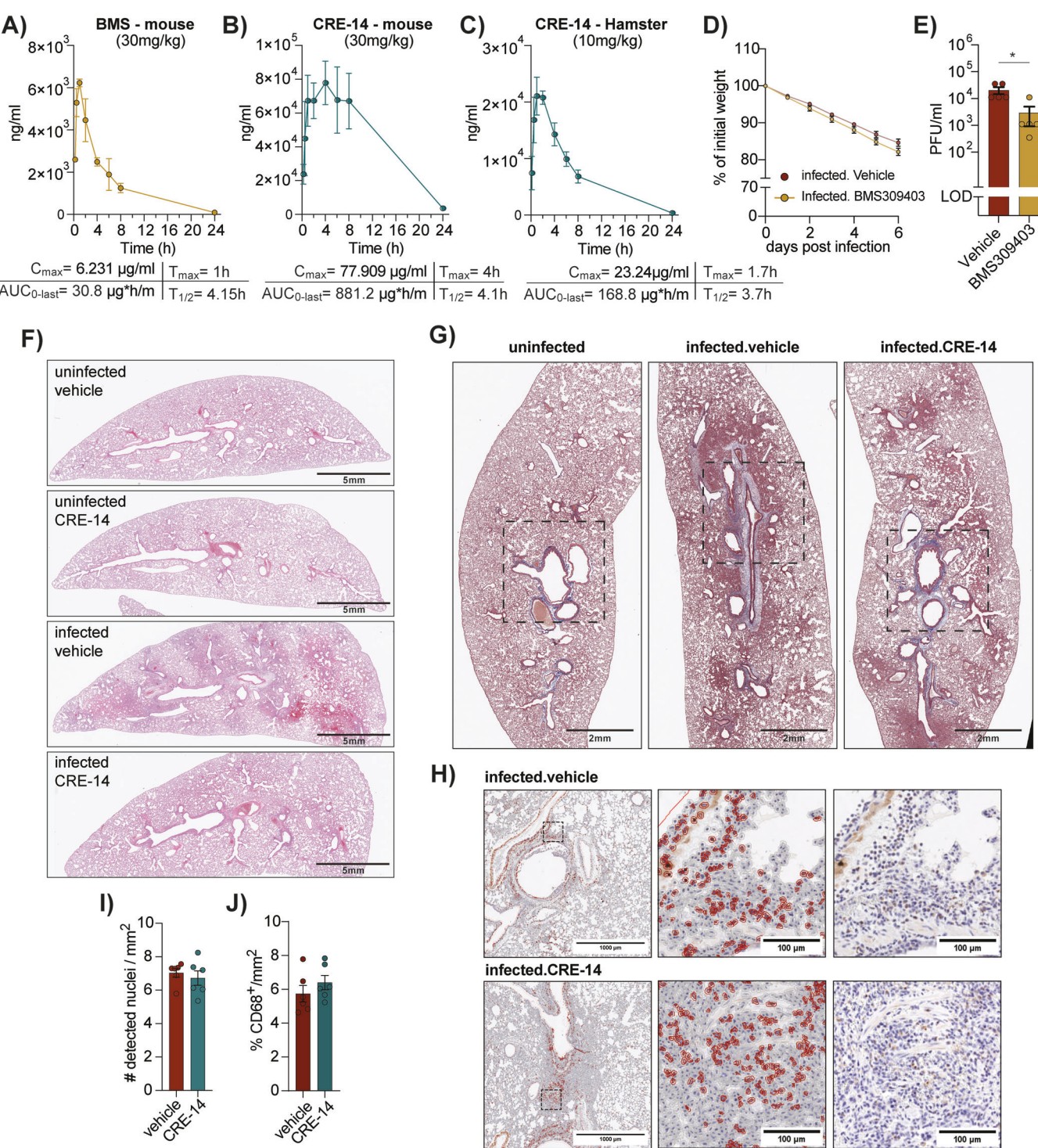

**Figure EV5. Pharmacokinetics of FABP4 inhibition in mice and hamsters.**

(A, B) Circulating concentrations of BMS309403 and CRE-14 following a 30 mg/kg subcutaneous injection in C57BL/6J mice. (C) Circulating concentrations of CRE-14 in Syrian hamsters following a subcutaneous injection of 10 mg/kg. Tables show (Cmax) maximal concentration, (Tmax) time to reach maximal concentration, (T1/2) half-life, and (AUC0-last) area under the curve from time zero to the last quantifiable time point ($n = 3$, biological replicates). (D) Percent body weight over time and (E) lung viral titer of hamsters infected with SARS-CoV-2 (Ank1 strain, 100 TCID50) with or without BMS309403 treatment (30 mg/kg) ($n = 5$, biological replicates, $*p = 0.0215$). Statistical analysis was performed using a standard t-test. (F) Representative H&E staining (scale bar = 5 mm) and (G) Masson's trichrome staining (scale bar=2 mm) of lung sections from control and infected hamsters with or without CRE-14 treatment ($n = 4$). (H) Representative IHC CD68 staining of infected hamster lungs (Ank1-Dlt strain, 1000 TCID50) with or without CRE-14 treatment (15 mg/kg), shown at low (Scale bar = 1 mm) and high (Scale bar = 100 μm) magnification. The mid-panel highlights CD68-positive cells. (I) Number of detected cells normalized to lung tissue area per lung and (J) percentage of CD68-positive cells quantified from CD68 IHC staining represented in (H). (H–J: $n = 6$ lungs from 3 biological replicates – hamsters). Data are shown as mean ± s.e.m. Source data are available online for this figure.

