## [Peer Review File · EMBO Molecular Medicine]

FABP4 as a Therapeutic Host Target Controlling SARS-CoV2 Infection

Gokhan Hotamisligil, Hatoon Baazim, Emre Koyuncu, Gürol Tuncman, M. Burak, Lea Merkel, Nadine Bahour, Ezgi Karabulut, Yankun Lee, Alireza Hanifehnezhad, Zehra Karagoz, Katalin Földes, Ilayda Engin, Ayse Erman, Sidika Oztop, Nazlican Filazi, Buket Gul, Ahmet Ceylan, Ozge Cinar, Fusun Can, Hahn Kim, Ali Al-Hakeem, Hui Li, Fatih Semerci, Xihong Lin, Erkan Yilmaz, Onder Ergonul, and Aykut Ozkul

Corresponding authors: Gokhan Hotamisligil (ghotamis@hsph.harvard.edu) , Aykut Ozkul (ozkul@ankara.edu.tr)

Review Timeline:

Submission Date:	9th May 24
Editorial Decision:	26th Jun 24
Revision Received:	22nd Nov 24
Editorial Decision:	5th Dec 24
Revision Received:	16th Dec 24
Accepted:	17th Dec 24

Editor: Zeljko Durdevic

Transaction Report:

26th Jun 2024

Dear Prof. Hotamisligil,

Thank you for the submission of your manuscript to EMBO Molecular Medicine, and please accept my apologies for the delay in getting back to you as we were waiting for one referee report. However, given that referee #3 has not yet gotten back to us despite several chasers, and that both referees #1 and #2 provide similar recommendations, we prefer to make a decision now in order to avoid further delay in the process. As you will see from the reports below, the referees acknowledge the interest of the study and are overall supporting publication of your work pending appropriate revisions.

Addressing the reviewers' concerns in full will be necessary for further considering the manuscript in our journal, and acceptance of the manuscript will entail a second round of review. EMBO Molecular Medicine encourages a single round of revision only and therefore, acceptance or rejection of the manuscript will depend on the completeness of your responses included in the next, final version of the manuscript. For this reason, and to save you from any frustrations in the end, I would strongly advise against returning an incomplete revision.

I look forward to seeing a revised form of your manuscript as soon as possible.

Yours sincerely,

Zeljko Durdevic

We require:

- 1) A .docx formatted version of the manuscript text (including legends for main figures, EV figures and tables). Please make sure that the changes are highlighted to be clearly visible.
- 2) Individual production quality figure files as .eps, .tif, .jpg (one file per figure). For guidance, download the 'Figure Guide PDF': (<https://www.embopress.org/page/journal/17574684/authorguide#figureformat>).
- 3) A .docx formatted letter INCLUDING the reviewers' reports and your detailed point-by-point responses to their comments. As part of the EMBO Press transparent editorial process, the point-by-point response is part of the Review Process File (RPF), which will be published alongside your paper.
- 4) A complete author checklist, which you can download from our author guidelines (<https://www.embopress.org/page/journal/17574684/authorguide#submissionofrevisions>). Please insert information in the checklist that is also reflected in the manuscript. The completed author checklist will also be part of the RPF.
- 5) Please note that all corresponding authors are required to supply an ORCID ID for their name upon submission of a revised manuscript.
- 6) It is mandatory to include a 'Data Availability' section after the Materials and Methods. Before submitting your revision, primary datasets produced in this study need to be deposited in an appropriate public database, and the accession numbers and database listed under 'Data Availability'. Please remember to provide a reviewer password if the datasets are not yet public (see <https://www.embopress.org/page/journal/17574684/authorguide#dataavailability>).

In case you have no data that requires deposition in a public database, please state so in this section. Note that the Data

Availability Section is restricted to new primary data that are part of this study.

13) Author contributions: You will be asked to provide CRediT (Contributor Role Taxonomy) terms in the submission system. These replace a narrative author contribution section in the manuscript.

14) A Conflict of Interest statement should be provided in the main text.

15) Every published paper now includes a 'Synopsis' to further enhance discoverability. Synopses are displayed on the journal webpage and are freely accessible to all readers. They include a short stand first (maximum of 300 characters, including space) as well as 2-5 one-sentences bullet points that summarizes the paper. Please write the bullet points to summarize the key NEW findings. They should be designed to be complementary to the abstract - i.e. not repeat the same text. We encourage inclusion of key acronyms and quantitative information (maximum of 30 words / bullet point). Please use the passive voice. Please attach

these in a separate file or send them by email, we will incorporate them accordingly.

16) Include a Reagents and Tools Table as part of the Methods section, which can be downloaded from our author guidelines (<https://www.embopress.org/page/journal/17574684/authorguide#structuredmethods>)

***** Reviewer's comments *****

Referee #1 (Comments on Novelty/Model System for Author):

The manuscript is highly novel and impactful as it identifies a cellular factor, FABP4, that may be a target for controlling SARS-CoV2 infection. The broad applicability and usefulness of FABP4 inhibitors may be a therapeutic advantage previously not considered.

A major weakness of the study is the lack of attention to immune cells and this can be rectified by assessing CD45+ profiles.

Referee #1 (Remarks for Author):

This is a very interesting and potentially impactful paper by a very strong group with a history of groundbreaking studies on FABP4. Baazim et al. report that genetic or pharmacological loss of FABP4 results in reduced viral titre and loss of viral replication organelles in human cell lines and organoids suggesting FABP4 as a therapeutic target for SARS-CoV2 infection. In a hamster model, FABP4 inhibitors attenuated lung damage and reduced viral load. Overall, the work is done very well and is high impact. A strength of the study is the use of several viral variants and not the reliance on a single variant subtype making the conclusions likely more broadly applicable. During review, several concerns arose concerning conclusions and controls and these are details in comments to the authors.

A. There is an almost complete absence of mention or analysis of immune cell composition in the presence or absence of virus and the impact of FABP4 loss. This group, plus others, have reported that FABP4 is also found in macrophages and that immune cell loss of FABP4 drives cardiometabolic improvement. As such, the question arises as to whether loss of FABP4 in macrophages, or the balance between M1/M2 macrophages affects viral properties in adipocytes or other cells. Have the authors profiled macrophages in their system (particularly the hamster system) and if so, how does loss of FABP4 affect CD45+ immune cell profiles?

B. The authors conclude that replication is potentiated by lipid droplets but that infection results in depletion of such lipid stores (5/15). Have the authors profiled expression of genes linked to either DNL or lipolysis in such infected cells. Is the loss of lipid droplets reduced biosynthesis or enhanced lipolysis? Can there be information gleaned from evaluating the expression of metabolic enzyme mRNA levels?

C. In the hamster model system, following infection the authors indicate that pharmacologic inhibition of FABP4 results in attenuated infectivity. This is a conclusion based largely on a single time point. If the process is followed longitudinally, do inhibitor cells acquire the same viral load, albeit at a later time. That is, does the FABP4 inhibitor prevent or delay infectivity and viral sequelae?

Minor comments:

- a. Extended data 2. Typo panels E and G labeling
- b. Extended data 2, what is the difference between panels K and L? This should be clarified.

Referee #2 (Comments on Novelty/Model System for Author):

The authors address the role of FABP4 in various mammalian cell lines, as well as in a hamster model in vivo. Furthermore, the authors examine FABP4 in human clinical isolates. Collectively, the models used indicate an important role for FABP4 in viral pathogenesis during COVID-19 infection.

Referee #2 (Remarks for Author):

Most infections lead to drastic changes in cellular and organismal metabolism. Yet, we have little mechanistic insight into the

role of metabolism in the progression of infectious disease, a topic that has far-reaching implications for human health. Here, Baazim et al investigate the role of fatty acid-binding protein 4 (FABP4) during in cellular and animal models of SARS-CoV-2 infection. FABP4, which the authors show is correlated with disease severity in COVID-19 patients, is recruited to DMVs and promotes viral replication in various cellular models. The genetic ablation or pharmacological inhibition of FABP4 impairs viral replication in several in vitro models, highlighting its potential as a therapeutic target. The authors test this latter point and find that FABP4 inhibition decreased viral titers in lungs of infected hamsters. In summary, this elegant and comprehensive study highlights a key host vulnerability amenable to therapeutic targeting for not only for SARS-CoV-2 but other viruses, as well as present a potential model by which FABP4 regulates SARS-CoV-2 infection. I just have a few follow-up points for clarification:

How does FABP4 affect viral replication? The loss of FABP4 affects viral replication; given that FABP4 localizes to regions of viral replication, the authors speculate this is due to FABP4 promoting viral access to host lipids. However, an alternative explanation is that FABP4 simply reduces LDs stores in cells. The authors should address this by examining the effect of FABP4 on LD number and size in uninfected cells.

Correlation between FABP4 and SARS-CoV-2 infection? It is interesting that FABP4 levels are increased in lungs of COVID-19 patients given that in Fig. 2F and 3L, infection seems to lead to a decrease in FABP4. The authors should elaborate on this in the discussion.

FABP4 recruitment/colocalization with DMV. The authors show an overlap between FABP4 and dsRNA; however it is unclear if this is simply due to FABP4 localizing to the ER. To address this, the authors should have an IFA panel of dsRNA/FABP4/calnexin (or any other ER marker). If it is the latter, the authors should reassess their interpretation.

Off target effects of FABP4 inhibition: the authors should test whether the FABP4 inhibitor (for examples used in 4A-D) affects viral replication in FABP4 KO cells

Microscopy images: insets should be higher magnification to visualize overlap (i.e. subcellular rather than several cells) and individual panels should be in gray scale

Point to point response to reviewers.

We sincerely appreciate the reviewers thoughtful and constructive comments, which have helped improve the quality and clarify of our manuscript. Below, we address each of the comments in details. We have presented each of reviewer's comments in italicized quotes and responded below.

Referee #1

"The manuscript is highly novel and impactful as it identifies a cellular factor, FABP4, that may be a target for controlling SARS-CoV2 infection. The broad applicability and usefulness of FABP4 inhibitors may be a therapeutic advantage previously not considered.

A major weakness of the study is the lack of attention to immune cells and this can be rectified by assessing CD45+ profiles.

We thank the reviewer for these comments.

Referee #1 (Remarks for Author):

This is a very interesting and potentially impactful paper by a very strong group with a history of groundbreaking studies on FABP4. Baazim et al. report that genetic or pharmacological loss of FABP4 results in reduced viral titre and loss of viral replication organelles in human cell lines and organoids suggesting FABP4 as a therapeutic target for SARS-CoV2 infection. In a hamster model, FABP4 inhibitors attenuated lung damage and reduced viral load. Overall, the work is done very well and is high impact. A strength of the study is the use of several viral variants and not the reliance on a single variant subtype making the conclusions likely more broadly applicable.

We thank the reviewer for careful examination of our paper and for these encouraging positive remarks.

During review, several concerns arose concerning conclusions and controls and these are details in comments to the authors."

"A. There is an almost complete absence of mention or analysis of immune cell composition in the presence or absence of virus and the impact of FABP4 loss. This group, plus others, have reported that FABP4 is also found in macrophages and that immune cell loss of FABP4 drives cardiometabolic improvement. As such, the question arises as to whether loss of FABP4 in macrophages, or the balance between M1/M2 macrophages affects viral properties in adipocytes or other cells. Have the authors profiled macrophages in their system (particularly the hamster system) and if so, how does loss of FABP4 affect CD45+ immune cell profiles?"

We thank the reviewer for these comments. Indeed, we agree and do acknowledge that FABP4's immunomodulatory functions may play a role in the pathophysiology of SARS-CoV2 infection, both in terms of immune cell recruitment, and promoting M1-polarization in macrophages. However, we were unable to pursue these investigations for this manuscript due to the considerable logistical challenges of conducting experiments in the hamster model, particularly in a biosafety level 3 setting.

We did perform a new infection in hamsters and attempted to more comprehensively characterize changes in the immune cell population using flow cytometry. Regrettably, this analysis was not successful due to the lack of reagents compatible with hamster experiments and the failure of the mouse-targeting antibodies to detect hamster antigens for the F4/80 staining in the CD45+ population. We will plan these experiments in the future using mouse models of infection where the reagents are better characterized and validated.

In an effort to address the reviewer's questions, we performed IHC staining with CD68 to label macrophages in the lungs of hamsters infected with 1000 TCID50 of the Ank-Dlt strain, with or without CRE-14 treatment (15 mg/kg). Using QuPath's threshold-based cell detection, we quantified the number of nuclei per lung and identified CD68+ cells with a machine learning classifier trained on manually selected CD68-positive and CD68-negative cells. This allowed us to calculate the percentage of CD68+ cells relative to the total cell count, normalized to tissue area (mm²). Representative of the IHC images as well as the quantification results are now shown in (Figures EV. 5H-J). There are limitations of the conclusions that can be drawn from these experiments and a detailed characterization would be an interesting line to pursue in the future, in better characterized systems.

We also modified our discussion section to include an overview of the role of FABP4 in modulating cytokine secretion in a variety of cell types, and its regulation of macrophage and T cell function [Please refer to page 10, lines 11-38] which may relate to SARS-CoV2 infection.

“B. The authors conclude that replication is potentiated by lipid droplets but that infection results in depletion of such lipid stores (5/15). Have the authors profiled expression of genes linked to either DNL or lipolysis in such infected cells. Is the loss of lipid droplets reduced biosynthesis or enhanced lipolysis? Can there be information gleaned from evaluating the expression of metabolic enzyme mRNA levels?”

As per the reviewer’s suggestion, we examined the changes in lipolysis and lipogenesis in infected adipocytes by measuring the protein abundance of ATGL, HSL and phospho-HSL, and examining the expression of several genes involved in *de-novo* lipogenesis, including ACLY, ACC, FASN, SCD1 and DGAT1. We detected no changes in any of the measured parameters, which we suspect is influenced by the low proportion of infected adipocytes relative to the total number of cells, as indicated by the nucleocapsid positive staining in differentiated adipocytes (Figure EV2I). We would be happy to include this data in the manuscript at the reviewer’s request.

We added more details in the discussion to elaborate on the roles of lipolysis and de novo lipogenesis in SARS-CoV-2 replication. [Please refer to page 9, lines 24-30]

“C. In the hamster model system, following infection the authors indicate that pharmacologic inhibition of FABP4 results in attenuated infectivity. This is a conclusion based largely on a single time point. If the process is followed longitudinally, do inhibitor cells acquire the same viral load, albeit at a later time. That is, does the FABP4 inhibitor prevent or delay infectivity and viral sequelae?”

While we do agree with that a longitudinal analysis of viral titers would be informative, we were unfortunately unable to conduct further hamster experiments due to the significant logistical limitations associated with this model. We hope to conduct such analysis in future studies using the mouse models of infection, where our repertoire of tools to examine FABP4 biology is much more expansive.

“Minor comments:

a. Extended data 2. Typo panels E and G labeling

b. Extended data 2, what is the difference between panels K and L? This should be clarified.”

Thank you for pointing this out, both points have been addressed in the figure legends.

Referee #2:

“(Comments on Novelty/Model System for Author): The authors address the role of FABP4 in various mammalian cell lines, as well as in a hamster model in vivo. Furthermore, the authors examine FABP4 in human clinical isolates. Collectively, the models used indicate an important role for FABP4 in viral pathogenesis during COVID-19 infection.”

Referee #2 (Remarks for Author):

Most infections lead to drastic changes in cellular and organismal metabolism. Yet, we have little mechanistic insight into the role of metabolism in the progression of infectious disease, a topic that has far-reaching implications for human health. Here, Baazim et al investigate the role of fatty acid-binding protein 4 (FABP4) during in cellular and animal models of SARS-CoV-2 infection. FABP4, which the authors show is correlated with disease severity in COVID-19 patients, is recruited to DMVs and promotes viral replication in various cellular models. The genetic ablation or pharmacological inhibition of FABP4 impairs viral replication in several in vitro models, highlighting its potential as a therapeutic target. The authors test this latter point and find that FABP4 inhibition decreased viral titers in lungs of infected hamsters. In summary, this elegant and comprehensive study highlights a key host vulnerability amenable to therapeutic targeting for not only for SARS-CoV-2 but other viruses, as well as present a potential model by which FABP4 regulates SARS-CoV-2 infection. I just have a few follow-up points for clarification:

- 1. How does FABP4 affect viral replication? The loss of FABP4 affects viral replication; given that FABP4 localizes to regions of viral replication, the authors speculate this is due to FABP4 promoting viral access to host lipids. However, an alternative explanation is that FABP4 simply reduces LDs stores in cells. The authors should address this by examining the effect of FABP4 on LD number and size in uninfected cells.”*

We'd like to thank the reviewer for their kind and encouraging remarks. To address the reviewer's question, we've measured changes in the % LD area per cell across control and FABP4-inhibitor treated cells in the presence or absence of SARS-CoV2 infection (Figure EV. 3N). This analysis confirmed that FABP4 inhibition had no effect on the LD content neither in

the infected or uninfected cells. For the FABP4 knockout adipocyte model, we included a representative image of uninfected control and FABP4^{-/-} adipocytes post differentiation stained with Bodipy which showed that FABP4^{-/-} adipocytes are fully capable of differentiation and LD accumulation (Figure EV3I).

To further delineate the contribution FABP4 and LD accumulation to viral replication, we infected control and FABP4^{-/-} adipocytes at various stages of differentiation and measured the viral titers in the supernatant. This revealed that although viral titers were significantly reduced in the absence of FABP4, differentiating adipocytes still increased their capacity to replicate the virus, suggesting that both FABP4 and LDs can independently contribute to viral replication. This data is now integrated into (Figures EV3K-M), and discussed in page 9, lines 35-40.

“2. Correlation between FABP4 and SARS-CoV-2 infection? It is interesting that FABP4 levels are increased in lungs of COVID-19 patients given that in Fig. 2F and 3L, infection seems to lead to a decrease in FABP4. The authors should elaborate on this in the discussion. FABP4 recruitment/localization with DMV.”

We’ve examined the regulation of FABP4 abundance and expression during infection in (Figures EV2E,F) and found no changes in these parameters. We included a quantification of FABP4 band intensity relative to total proteins using Ponceau S staining to prevent potential complications

related to changes in housekeeping gene expression. This analysis confirmed that total intracellular FABP4 levels do not change in the whole cell lysate (Figure EV2F).

We then repeated the western blots for the samples shown in Figure 3 and the independent experiments in which the data was reproduced and confirmed that FABP4 levels remained

unchanged. (Figure 3F).

In our discussion section, we've speculated on the potential explanation for why we see no changes in adipocyte FABP4 intracellular or secreted levels in vitro, while circulating FABP4 in COVID-19 patients increased with disease severity. [Please see page 11, lines 9 to 19].

“3. The authors show an overlap between FABP4 and dsRNA; however it is unclear if this is simply due to FABP4 localizing to the ER. To address this, the authors should have an IFA panel of dsRNA/FABP4/calnexin (or any other ER marker). If it is the latter, the authors should reassess their interpretation.”

We re-analyzed the colocalization of FABP4 with the dsRNA and calnexin in samples stained for all three signals and included earlier timepoints to increase the temporal resolution. This analysis confirmed that in uninfected cells FABP4 is broadly distributed across the cytoplasm and does not colocalize with calnexin. Upon infection, FABP4 starts to colocalize with both dsRNA and calnexin as early as 8 hours post infection, with this colocalization peaking at 12 and 24 hours, while the colocalization between calnexin and dsRNA continued to increase over time (Figures 2L-P). Interestingly, when we examined active FABP4 secretion following a 1-hour incubation, we detected a reduction in secreted FABP4 12 hours post infection (Figure EV2H). Together this data suggests that FABP4 is retained within infected cells at the early stages of infection to support viral replication organelle biogenesis.

“4. Off target effects of FABP4 inhibition: the authors should test whether the FABP4 inhibitor (for examples used in 4A-D) affects viral replication in FABP4 KO cells”

As per the reviewer’s suggestion, we treated infected WT and FABP4^{-/-} adipocytes with varying concentrations of each inhibitor, then examined virus nucleocapsid levels 48 hours post infection from the cell lysates. This analysis revealed a dose-dependent reduction in nucleocapsid protein levels only in the WT infected cells, while in FABP4^{-/-} adipocytes, viral titers remained equally suppressed, further confirming that the effect of the inhibitor treatment on viral replication was a direct result of FABP4 inhibition. This data has been integrated into (Figure EV3K-M)

“Microscopy images: insets should be higher magnification to visualize overlap (i.e. subcellular rather than several cells) and individual panels should be in gray scale.”

We increased the magnification in all fluorescence microscopy images to focus on single large adipocytes or 2-3 smaller ones. Insets now highlight virus replication organelles at higher magnification, particularly in Figures 2L-M and 3J-K. We presented individual channels in grayscale, except in Figure 2M, where colors were retained to show overlap across three channels.

5th Dec 2024

Dear Prof. Hotamisligil,

Thank you for the submission of your revised manuscript to EMBO Molecular Medicine. I am pleased to inform you that we will be able to accept your manuscript pending the following final amendments:

- 1) Figures: Please indicate the magnification areas in Figure 4F. Also, indicate in the figure legends that images in Figure 5F and M are also presented in Figure EV5F and G.
- 2) In the main manuscript file, please do the following:
 - Please address all comments suggested by our data editors listed below:
 - o Figure legends:
 1. Please define the annotated p values ****/***/**/* as well as provide the exact p-values for the same in the legend of figure 1b-h; 2a-e, g-i, n-p; 3a-b, d-e, g-i, l-o; 4a-e, g-h; 5a-b, d, g-l; EV 1a-i; EV 2a-d, h; EV 3a, e-f, h, j, l-n, p; EV 4d; EV 5e; as appropriate.
 2. Please note that information related to n is missing in the legends of figures 1b-h; 5g-l; EV 1a-i; EV 3d-f, l-n.
 3. Although 'n' is provided, please describe the nature of entity for 'n' in the legends of figures 4e; EV 3a; EV 5i-j.
 4. Please note that scale bar and its definition are missing for figures EV 5f-g.
 - Correct order of manuscript sections: Abstract, Keywords, Introduction, Results, Discussion, Methods, Acknowledgements, Disclosure and competing interests statement, References, Figure legends, Tables and their legends, Expanded View Figure legends.
 - Rename "Materials and Methods" to "Methods".
 - Add up to 5 keywords.
 - In "Disclosure and competing interests statement" add the sentence: "Gökhan S. Hotamisligi is a member of the journal's advisory editorial board. This has no bearing on the editorial consideration of this article for publication."
 - Author contributions: Please remove it from the manuscript and specify author contributions in our submission system. CRediT has replaced the traditional author contributions section because it offers a systematic machine-readable author contributions format that allows for more effective research assessment. Please use the free text boxes beneath each contributing author's name to add specific details on the author's contribution. More information is available in our guide to authors: <https://www.embopress.org/page/journal/17574684/authorguide#authorshipguidelines>
 - In Methods, provide the antibody dilutions that were used for each antibody.
 - In Methods, please specify the biosafety level for the experiments with SARS-CoV-2 by adding and amending the following sentence: All experiments with SARS-CoV-2 were performed in a ... level laboratory and with approval from...
 - Indicate in legends number and nature of replicates and exact p= values, not a range, along with the statistical test used. To keep the figures "clear" some authors found providing an Appendix table Sx with all exact p-values preferable. You are welcome to do this if you want to.
 - Merge Funding with Acknowledgements. If project numbers are available, please added them to the manuscript text and to our system.
 - Move data availability section to the end of Methods and remove the sentence "Source data for the human cohorts and the histopathology evaluation are provided in the Expanded view tables (Appendix Tables S1-4)."
 - Please remove "Clinical management of COVID-19" form References.
- 3) Tables: Please rename Appendix/Supplementary Tables 2 and 3 to Dataset EV1 and Dataset EV2. Both need short legends added to the excel files in a separate tab/worksheet. Appendix/Supplementary tables 1 and 4 and Expanded View Content should be renamed Table EV1, Table EV2 and Table EV3, and they also need short legends added to the top of each page. Please update the callouts of the tables in the main text.
- 4) Source data: Please zip all EV Figure source data and upload them as one file.
- 5) Synopsis:
 - Synopsis image: Please resize the image to 550 px-wide x (300-600) px-high and upload it as high-resolution jpeg file.
 - Please check your synopsis text and image before submission with your revised manuscript. Please be aware that in the proof stage minor corrections only are allowed (e.g., typos).
- 6) As part of the EMBO Publications transparent editorial process initiative (see our Editorial at <http://embomolmed.embopress.org/content/2/9/329>), EMBO Molecular Medicine will publish online a Review Process File (RPF) to accompany accepted manuscripts. This file will be published in conjunction with your paper and will include the anonymous referee reports, your point-by-point response and all pertinent correspondence relating to the manuscript. Let us know whether you agree with the publication of the RPF and as here, if you want to remove or not any figures from it prior to publication. Please note that the Authors checklist will be published at the end of the RPF.
- 7) Please provide a point-by-point letter INCLUDING my comments as well as the reviewer's reports and your detailed responses (as Word file).

I look forward to receiving a new revised version of your manuscript as soon as possible.

Yours sincerely,

Zeljko Durdevic

*** Instructions to submit your revised manuscript ***

To submit your manuscript, please follow this link:

<https://embomolmed.msubmit.net/cgi-bin/main.plex>

- 1) a .docx formatted version of the manuscript text (including Figure legends and tables)
- 2) Separate figure files*
- 3) supplemental information as Expanded View and/or Appendix. Please carefully check the authors guidelines for formatting Expanded view and Appendix figures and tables at <https://www.embopress.org/page/journal/17574684/authorguide#expandedview>
- 4) a letter INCLUDING the reviewer's reports and your detailed responses to their comments (as Word file).
- 5) The paper explained: EMBO Molecular Medicine articles are accompanied by a summary of the articles to emphasize the major findings in the paper and their medical implications for the non-specialist reader. Please provide a draft summary of your article highlighting
 - the medical issue you are addressing,
 - the results obtained and
 - their clinical impact.This may be edited to ensure that readers understand the significance and context of the research. Please refer to any of our published articles for an example.
- 6) Author contributions: the contribution of every author must be detailed in a separate section.
- 7) EMBO Molecular Medicine now requires a complete author checklist (<https://www.embopress.org/page/journal/17574684/authorguide>) to be submitted with all revised manuscripts. Please use the checklist as guideline for the sort of information we need WITHIN the manuscript. The checklist should only be filled with page numbers where the information can be found. This is particularly important for animal reporting, antibody dilutions (missing) and exact values and n that should be indicated instead of a range.
- 8) Every published paper now includes a 'Synopsis' to further enhance discoverability. Synopses are displayed on the journal webpage and are freely accessible to all readers. They include a short stand first (maximum of 300 characters, including space) as well as 2-5 one sentence bullet points that summarise the paper. Please write the bullet points to summarise the key NEW findings. They should be designed to be complementary to the abstract - i.e. not repeat the same text. We encourage inclusion of key acronyms and quantitative information (maximum of 30 words / bullet point). Please use the passive voice. Please attach these in a separate file or send them by email, we will incorporate them accordingly.

You are also welcome to suggest a striking image or visual abstract to illustrate your article. If you do please provide a jpeg file 550 px-wide x 300-600px high.

9) A Conflict of Interest statement should be provided in the main text

10) Please note that we now mandate that all corresponding authors list an ORCID digital identifier. This takes <90 seconds to complete. We encourage all authors to supply an ORCID identifier, which will be linked to their name for unambiguous name identification.

Currently, our records indicate that the ORCID for your account is 0000-0003-2906-1897.

Link Not Available

11) Include a Reagents and Tools Table as part of the Methods section, which can be downloaded from our author guidelines (<https://www.embopress.org/page/journal/17574684/authorguide#structuredmethods>)

Photos 400-800 DPI

*Additional important information regarding figures and illustrations can be found at

<https://bit.ly/EMBOPressFigurePreparationGuideline>. See also figure legend preparation guidelines:

<https://www.embopress.org/page/journal/17574684/authorguide#figureformat>

***** Reviewer's comments *****

Referee #1 (Comments on Novelty/Model System for Author):

No ethical issues with human and hamster models, all work approved. Impact is high due to health implications and opportunity to develop novel therapeutic.

This revised manuscript by Baazim is impactful and potentially important providing a new therapeutic approach towards SARS and SARS-type viral infections. The PI and team have been extremely responsive to the prior review and included new information in the paper that strengthens the report. In my prior review I asked primarily about immune cell populations and in this revised manuscript, the authors explain their workflow and the experiments they performed. They also included new text in the discussion. Secondly, I asked about DNL and lipolysis and the authors provide new information on expression of target genes related to lipid droplets, etc.

Reviewer #2 made insightful comments as well and the sum changes due to both reviews strengthens this paper.

Overall, the paper is much improved and I recommend acceptance.

Referee #1 (Remarks for Author):

accept

The authors addressed the remaining editorial issues.

17th Dec 2024

Dear Prof. Hotamisligil,

We are pleased to inform you that your manuscript is accepted for publication and is now being sent to our publisher to be included in the next available issue of EMBO Molecular Medicine.
